# DeMAG predicts the effects of variants in clinically actionable genes by integrating structural and evolutionary epistatic features

Federica Luppino[1,2], Ivan A. Adzhubei[3,4], Christopher A. Cassa [3] ✉ &
Agnes Toth-Petroczy [1,2,5] ✉

Despite the increasing use of genomic sequencing in clinical practice, the interpretation of rare genetic variants remains challenging even in well-studied disease genes, resulting in many patients with Variants of Uncertain Significance (VUSs). Computational Variant Effect Predictors (VEPs) provide valuable evidence in variant assessment, but they are prone to misclassifying benign variants, contributing to false positives. Here, we develop Deciphering Mutations in Actionable Genes (DeMAG), a supervised classifier for missense variants trained using extensive diagnostic data available in 59 actionable disease genes (American College of Medical Genetics and Genomics Secondary Findings v2.0, ACMG SF v2.0). DeMAG improves performance over existing VEPs by reaching balanced specificity (82%) and sensitivity (94%) on clinical data, and includes a novel epistatic feature, the 'partners score', which leverages evolutionary and structural partnerships of residues. The 'partners score' provides a general framework for modeling epistatic interactions, integrating both clinical and functional information. We provide our tool and predictions for all missense variants in 316 clinically actionable disease genes (demag.org) to facilitate the interpretation of variants and improve clinical decision-making.

Assessing the pathogenicity of genetic variants remains a significant challenge in research and clinical translation. The American College of Medical Genetics and Genomics (ACMG) recommends the reporting of secondary findings in clinically actionable genes (e.g., ACMG SF lists[1,2]) when patients undergo sequencing[3]. Knowledge of a pathogenic variant in such a gene might improve clinical management, diagnosis, and prevention. Given insufficient epidemiological, functional, or other supportive evidence, over three quarters of variants which have been submitted to ClinVar[4] are classified as Variants of Uncertain

Significance (VUSs, Supplementary Fig. 1). The uncertainty about the pathogenicity of a variant may pose a psychological burden for patients[5,6], left without guidance, and can lead to potential morbidity and health costs associated with under and overdiagnosis[7].

Many Variant Effect Predictors (VEPs) have been developed to predict the functional impacts of these variants, and these tools are often used in diagnostic variant interpretation[8–12]. A computational evidence that a variant is predicted to have a deleterious effect is considered "supporting evidence of pathogenicity" when following the

[1]Max Planck Institute of Molecular Cell Biology and Genetics, 01307 Dresden, Germany. [2]Center for Systems Biology Dresden, 01307 Dresden, Germany. [3]Brigham and Women's Hospital Division of Genetics, Harvard Medical School, Boston, MA 02115, USA. [4]Department of Biomedical Informatics, Harvard Medical School, Boston, MA 02115, USA. [5]Cluster of Excellence Physics of Life, TU Dresden, 01062 Dresden, Germany. ✉e-mail: ccassa@bwh.harvard.edu; toth-petroczy@mpi-cbg.de

American College of Medical Genetics and Genomics/Association for Molecular Pathology (ACMG/AMP) clinical guidelines for sequence variant interpretation[13,14]. Most commonly used VEPs are supervised methods which are trained using lists of pathogenic and benign variants, and assign variants a pathogenicity score using sequence-based and structural features. While most tools are designed to be used exome-wide, specialized predictors can reach higher performance on selected genes and disease phenotypes[15].

Unsupervised methods, such as DeepSequence[16], EVmutation[17], and EVE[18] are agnostic to variant labels as they infer functional effects from multiple sequence alignment (MSA). These methods rely on the availability of high quality MSA data, which is often missing in disordered and low-complexity regions, and poorly conserved regions[19]. Unsupervised methods characterize the fitness effects of mutations independently from reported disease-causing variants, and do not provide an interpretation of pathogenicity[17,20]. An exception is EVE, which provides two gene-specific unsupervised thresholds for pathogenic and benign variants respectively, however it leaves the most uncertain variants without annotation[18]. The method relies on labeled clinical data to identify the uncertain class. While this is useful for clinical applications, it suffers from labeling biases of supervised tools that use publicly available variants databases[21].

Due to limited clinical data, there are two primary challenges in training sufficiently accurate VEPs[21]. The first issue (type 1 circularity) refers to a biased testing set and requires that the testing set contains variants that were not used in the training of all supervised predictors. This is challenging as many methods train models using variants collected from similar sources, and can result in general inflation of predictive performance. The second issue (type 2 circularity) refers to an intrinsic characteristic of clinical databases: variants in a given gene, with an established link to a disease phenotype, may often be classified as pathogenic[22]. VEPs which use gene-based features, e.g., length of the protein, can make predictions based on a gene's characteristics and pathogenicity, rather than on the attributes of a specific variant. This bias hinders discrimination between pathogenic and benign variants within a given gene and skews the predictive performance toward high sensitivity and poor specificity[23]. Addressing these issues is crucial for the development of an accurate predictor for clinical applications.

Comparative genomics and 3D structures of proteins contain valuable information about the importance of residue positions and substitutions. The evolutionary conservation of a position in orthologous sequences correlates with the tolerance to mutations within a population, and can be used to predict the pathogenicity of genetic variants[24]. Several conservation scores have been developed and are used as predictive features in VEPs[24–26]. While most assume site-independence, considering epistasis between pairs of residue positions improves discrimination between disease-associated and common variants[17]. Here, epistasis refers to the interdependence of the two residue positions. An estimated 90% of variation is impacted by epistasis[27,28]. For example, ~10–15% of substitutions in non-human proteins are known to be pathogenic in their human orthologs[29,30]. These are termed compensated pathogenic deviations[29] as the pathogenicity of the substitution is suppressed by another compensatory substitution either within the same gene[31–35] or in another one[36]. The compensatory mechanism often involves residues in close proximity in the 3D structure and the preservation of side-chain to side-chain interactions[29]. In general, the hydrophobic core of proteins tends to evolve slowly, while the surface evolves more quickly[37]. Accordingly, disease-causing mutations tend to occur in the hydrophobic core of the 3D structure of the protein, while common variants tend to be located on the surface, i.e. areas with high solvent accessibility[38]. Incorporating epistatic and structural information comprehensively only recently became possible at large scale with the arrival of AlphaFold2[39] 3D protein structure predictions, because many disease-related genes do not have an experimentally resolved crystal structure.

Here, we extend the traditional conservation paradigm to assess variant effects with novel protein sequence- and structure-based features. We designed an epistatic feature, the partners score, which defines epistatic residue pairs based on co-evolutionary and 3D structural partnership of residues as defined by AlphaFold2[39] models. The partners score is informed by the clinical labels of partner residues, taking advantage of the wealth of existing clinical knowledge. Based on their medical importance and the abundance of clinical diagnostic data, we focused on interpreting missense variants in 59 clinically actionable disease genes in the ACMG SF v2.0 list, which we refer to as ACMG SF genes[2].

In this work, we develop DeMAG (Deciphering Mutations in Actionable Genes), a specialized supervised classifier for the 59 ACMG SF genes. DeMAG achieves the best overall performance across VEPs when tested on variants with clinical annotations. Further, we evaluate DeMAG on variants in additional 257 clinically associated disease genes, that our model has not been trained on, and found that it has high predictive power, reaching 91% sensitivity and 85% specificity. We anticipate that as additional clinical data becomes available, more genes can benefit from the partners score feature and from DeMAG predictions in general. We share predictions and interpretations of all ~1.3 million missense variants in the 59 ACMG SF genes and ~4.3 million variants for 257 other clinically relevant genes as a web application (demag.org) and provide our software and data for download (git.mpi-cbg.de/tothpetroczylab/demag/).

## Results
### Methodological overview
We developed DeMAG (Deciphering Mutations in Actionable Genes) a supervised classifier to assess the pathogenicity of mutations in 59 clinically actionable disease genes (ACMG SF v2.0 list) and support clinical decision making. First, we carefully curated pathogenic and benign variants used for training the model (Fig. 1 and Supplementary Fig. 2). For those variants, we then tested several sequence- and structure-based features and selected those that discriminated between variants with high confidence pathogenic and benign classifications (Fig. 1 and Supplementary Table 1). We designed the partners score, which is based on evolutionary and structural partnerships of residues as estimated by AlphaFold2 structural models (Figs. 2 and 3 and Supplementary Fig. 3). Overall, DeMAG used only 13 features, 8 derived from sequence conservation, and 5 from 3D structural models, disorder scores and epistatic relationships (Supplementary Table 2).

We trained a machine learning model (Fig. 4), and validated it with 3 different ground-truth test sets: clinical (Fig. 5 and Table 1), functional (deep mutational scanning, Supplementary Fig. 4), and benign variants from population data (Fig. 6 and Table 2). We further evaluated its performance on an additional set of 257 clinically relevant genes, which have sufficient numbers of variants with high quality diagnostic interpretations (Supplementary Table 4 and Supplementary Data 1). Finally, we computed DeMAG pathogenicity scores for all missense variants in the ACMG SF genes and additional 257 clinically relevant genes.

### Curated training set
In order to curate a high-quality training set, we considered several independent sources of SNVs and set strict criteria to retain only high-quality variants. We included high-quality ClinVar benign and pathogenic variants with a review status of at least 'two stars', namely variants labeled with no conflicts between all submitters or reviewed by expert panels or practice guidelines. We supplemented pathogenic variants with variants which have previously been described in the medical literature in the Human Gene Mutation Database (HGMD)[40], that have not yet been observed in ClinVar (Supplementary Fig. 5). The last source included all disease-causing mutations from UniProtKB. In

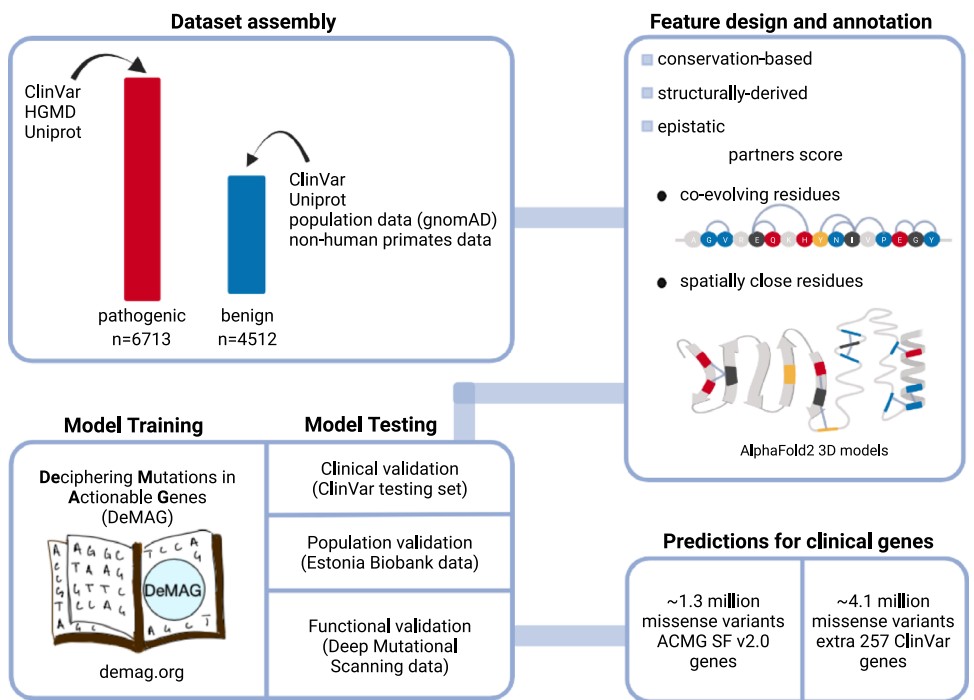

**Fig. 1 | Overview of DeMAG (Deciphering Mutations in Actionable Genes).** First, we assembled a training set from 59 actionable genes; pathogenic variants were collected from clinical databases such as ClinVar and Human Gene Mutation Database (HGMD[40]). The benign class includes variants from different sources including clinical, population, and non-human primate variants. The training set consisted of 6713 (60%) pathogenic and 4512 (40%) benign variants. Next, we annotated each variant with features, including EVmutation[18], IUPred2A[75], and AlphaFold2[40] confidence score (predicted Local Distance Difference Test, pLDDT). We designed a novel feature, the 'partners score', that captures the epistatic effects both in the sequence and in the 3D space of the protein. It relies on the observation that evolutionary coupled and spatially close residues are enriched in the same phenotypic effect. Next, we trained a classification tree-based gradient-boosting model using 13 selected features. The model was validated with 3 different types of data: (1) clinical testing data from ClinVar, (2) benign variants from the Estonian Biobank, and (3) deep mutational scanning data for four genes (*BRCA1, TP53, MSH2,* and *PTEN*). Finally, we provide predictions for all variants in the 59 actionable genes and extra 257 ClinVar genes and a web application (demag.org). Created with BioRender.com.

total, the pathogenic class consisted of 6713 unique pathogenic mutations (Fig. 1 and Supplementary Fig. 2a).

In addition to ClinVar, we collected benign variants from a large population database, gnomAD[41] (Genome Aggregation Database), and additional population-specific databases, including individuals of Korean[42] and Japanese[43] ancestry, as well as human orthologous polymorphisms[30] (Supplementary Fig. 2a). We defined benign variants as those with a minor allele frequency (MAF) greater than the associated disease prevalence; that is considered "Strong evidence of benign impact" according to ACMG-AMP guidelines[13,14]. Using a disease-specific MAF threshold, we gained almost 3000 benign variants compared to using a generic MAF > 0.5% threshold (Supplementary Fig. 2b). The benign class consisted of 4512 variants. The above approach of using gene-specific MAF thresholds can generally be applied to other genes to increase the number of benign variants. Overall, we have a relatively balanced training set, which includes 40% benign and 60% pathogenic variants (Supplementary Fig. 2d).

## Development of the partners score to incorporate epistatic effects

We designed a novel feature called the partners score based on the observation that partner residues that are connected, either because they are close in 3D proximity or because they are co-evolving, share the same phenotypic effect (Supplementary Fig. 6a). We used the AlphaFold2 3D protein structural models to identify residues in spatial proximity (<11 Å between C-alpha atoms, see Methods section) and highly correlated positions inferred from multiple sequence alignments of homologous sequences[44,45], to identify co-evolving residue pairs. Co-evolving residues account for 13% of all partnerships while spatially close ones account for the remaining 87%. There is an overlap

between the two types of partners, and 87% of co-evolving partners are also spatially close partners in 3D space of the protein (Supplementary Fig. 6c).

Each residue position can be associated with only pathogenic, only benign, both pathogenic and benign (mixed), or not being associated with any known variant (Fig. 2a). Each residue has a score (residue score) based on the type and number of connections it has (Fig. 2b). We used a mixture-based discriminant analysis[46] approach to define the partners score: first, the density of the residue score is estimated independently for the pathogenic and benign class in the training set assuming a gaussian mixture distribution (Fig. 2b). Then, each variant is assigned a posterior probability of belonging to either class, given the residue score and the prior probability of both classes (i.e., frequency). The posterior probability of pathogenicity defines the partners score (Fig. 2c and Methods section) which highlights how mutations with the same phenotypic effects cluster both in linear and 3D space of the protein.

## Partners score identifies functional sites

While only 13% of ACMG SF residues have annotated variants in the training set, we can inform 74% of positions with the partners score by making use of epistatic relationships (Fig. 2d). For example, the DNA mismatch repair protein MSH6 has only 8 pathogenic and 7 benign residue positions that are also co-evolving with other positions. With the partners score, we annotated 255 positions whose clinical significance has not been assessed yet. The same trend applies to positions in spatial proximity (Fig. 3a): amongst the spatially close residue positions, only 53 have annotated variants (33 pathogenic and 20 benign). With the partner score based on spatial proximity, we annotated 750 positions. Overall, if we consider both evolutionary and

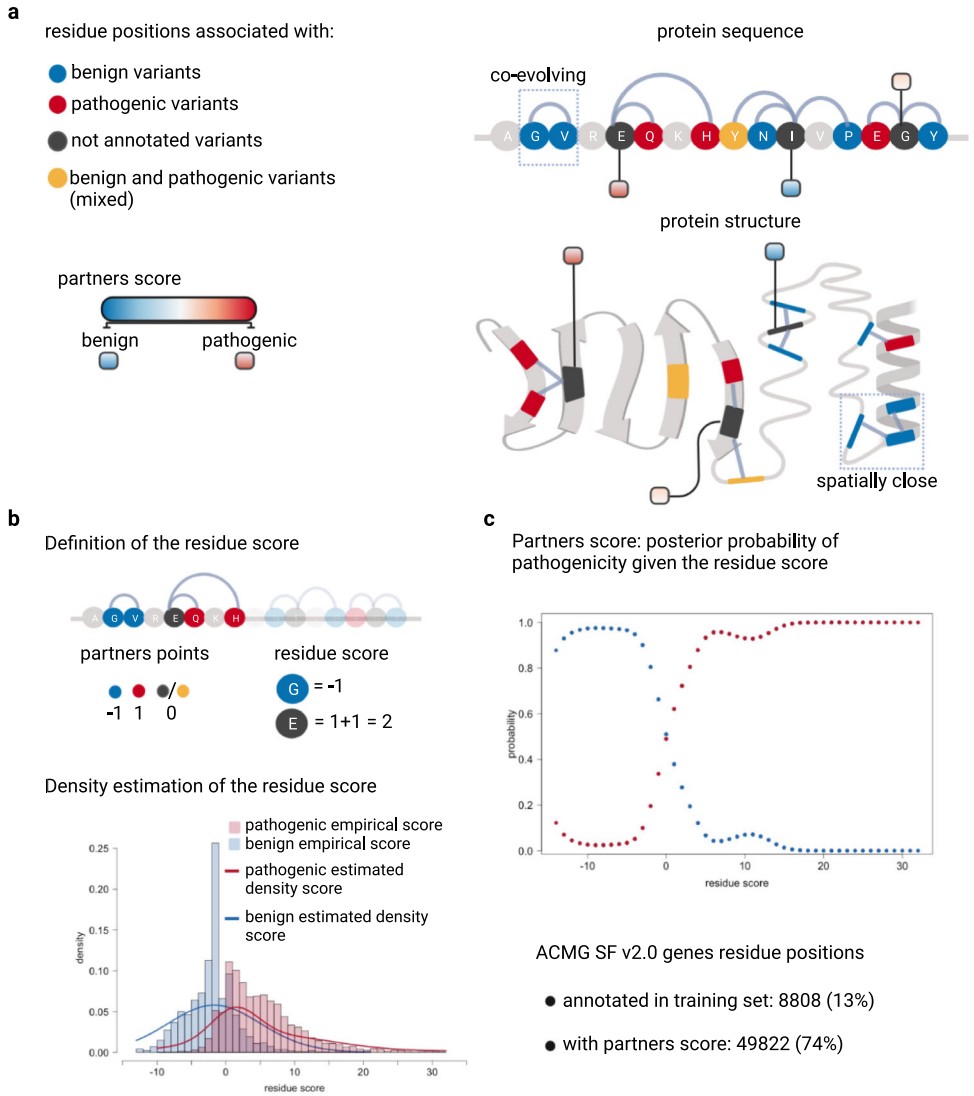

**Fig. 2 | The partners score – integrating evolutionary and structural information to inform variant assessment. a** Co-evolving residue pairs in the protein sequence and spatially adjacent residues (<11 Å) in the protein 3D structure. Residues are colored according to the phenotypic effect of associated variants within the DeMAG training set, e.g., red for positions with only pathogenic variants (see legend at left). The partners score informs residues positions that do not have any associated variants but are either co-evolving or spatially close to residue positions that have variants with known phenotypic effect (i.e., annotated in the training set). **b** A point scale quantifies the connection between residue positions e.g., benign positions are assigned −1, pathogenic 1, and mixed or not annotated 0. Next, we defined the residue score as the sum of partners points per residue e.g., residue glutamic acid (E) is connected with two pathogenic variants thus its residue score is 2. The partners score is derived via a mixture discriminant analysis model. First, the residue score distribution (histograms) for the benign and pathogenic residue positions is estimated with a gaussian mixture model (density lines). **c** The posterior probability of belonging both to the pathogenic and benign class is computed given the residue score and the prior probability (i.e., frequency of the two classes). The partners score is defined as the posterior probability of pathogenicity. The partners score feature is available for 74% of ACMG SF v2.0 residues positions, while only 13% of positions have annotations in the training set. Created with BioRender.com.

spatial partnerships, the partners feature assigns a score to 55% of all MSH6 residues (750 positions). For the cellular tumor protein P53, we observed a clear correlation between the partners score and Pfam[47] protein domain annotations, e.g., residue positions of the low-complexity region and disordered region are characterized by low partners scores, while residue positions of the DNA-binding domain has overall high scores (Fig. 3a and Supplementary Fig. 7). In addition, we observed that the MSH6 ATP binding site has a partners score >0.6 (Fig. 3b). The role of the ATP binding site of the MSH2-MSH6 hetero-dimer is crucial for DNA mismatch repair (MMR) competency: mutations of the lysine residue in the MSH6 Walker A motif are complete loss of function mutations in vivo in *S. cerevisiae*[48]. Moreover, all 14 mutations (G1134[A,R,E,V], P1135A, N1136D, M1137[T,V], G1138R, G1139[D,C,V], S1141[C,P]) in this site are ClinVar VUSs, with no

definitive clinical interpretation, while they are predicted pathogenic by DeMAG.

Overall, 67% of variants located in Pfam domains are pathogenic in our dataset. Additionally, we find that variants in Pfam domains have higher partners scores (Supplementary Fig. 7, *p*-value <2.2e-16) supporting the utility of this feature in assessing variant effects.

**DeMAG reaches high sensitivity and specificity**

Several existing VEPs, such as M-CAP and SIFT4G have high sensitivity but low specificity[23]. Their recommended thresholds (M-CAP[10] 0.025 and SIFT4G 0.05[24]) are set to reach high sensitivity in variant interpretation, while tolerating a high misclassification rate for benign variants. This imbalance increases the number of potentially false

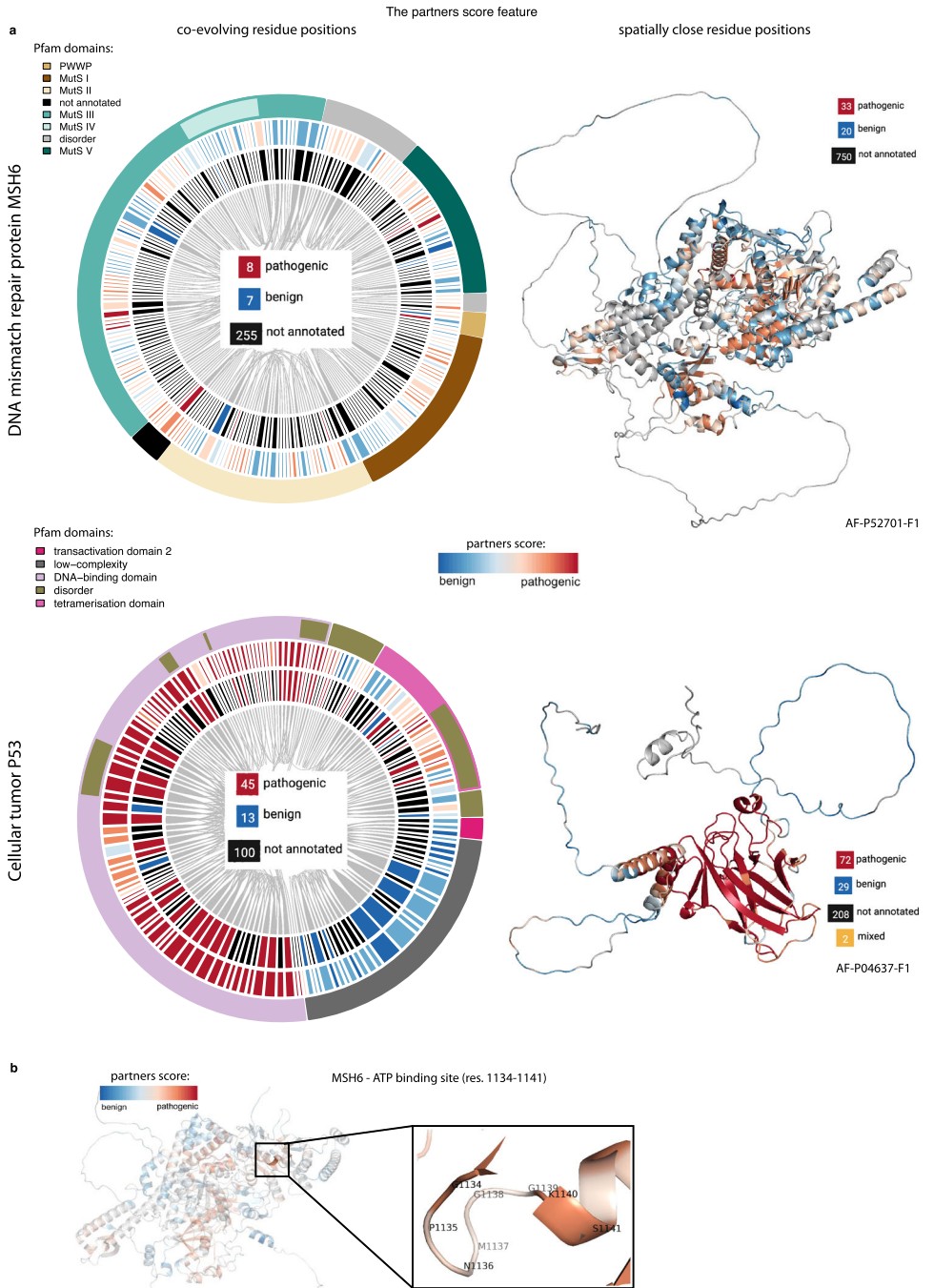

**Fig. 3 | The partners score annotates residue positions lacking prior functional or clinical assessments. a** On the top left, co-evolving residue positions are shown for the DNA mismatch repair protein MSH6. The innermost circle indicates residues that are co-evolving with at least one other residue whose phenotypic effect is known (i.e., annotated in the training set). Among such residues, 8 are pathogenic, 7 benign, and 255 lack prior assessment. The outer circle shows the partners score resulting from co-evolutionary partnership of residues, which gives a score to 255 residue positions without known annotations. The outermost circle indicates the Pfam domains of MSH6. On the right, AlphaFold 3D model of the protein is shown. Residue positions are colored based on the partners score derived from spatially close partnerships of residues, which inform 750 previously unannotated residue positions (55% of the protein length). Below, the same representation is shown for the cellular tumor antigen P53 protein. The circle plot shows correlation between partners score and domain annotation: the residues that belong to the DNA-binding domain have high partners scores while in the low-complexity region have low partners scores (Supplementary Fig. 7). In total 208 (53%) previously unlabeled residue positions are now annotated with partner scores. **b** A structural example, where MSH6 ATP binding site residues, have been previously shown[50] to be critical for DNA mismatch repair (MMR) function, have a high partners score.

positive variants (benign variants predicted incorrectly to be pathogenic). To address this issue, we made extensive efforts to improve training set balance by expanding the number of available benign mutations (Fig. 1 and Supplementary Fig. 2d). We selected only 13 features with balanced performance in discriminating between pathogenic and benign classes (Supplementary Table 1a and Methods section), including 8 derived from sequence conservation, and 5 from 3D structural models disorder scores, and epistatic relationships (Supplementary Table 2). DeMAG was trained with a gradient-boosting tree method[49,50] (see Methods section) and it yielded high accuracy (87%) and AUC-ROC (92%) values that correspond to high sensitivity (87%) and specificity (85%), as well as high precision (90%) (Fig. 4c).

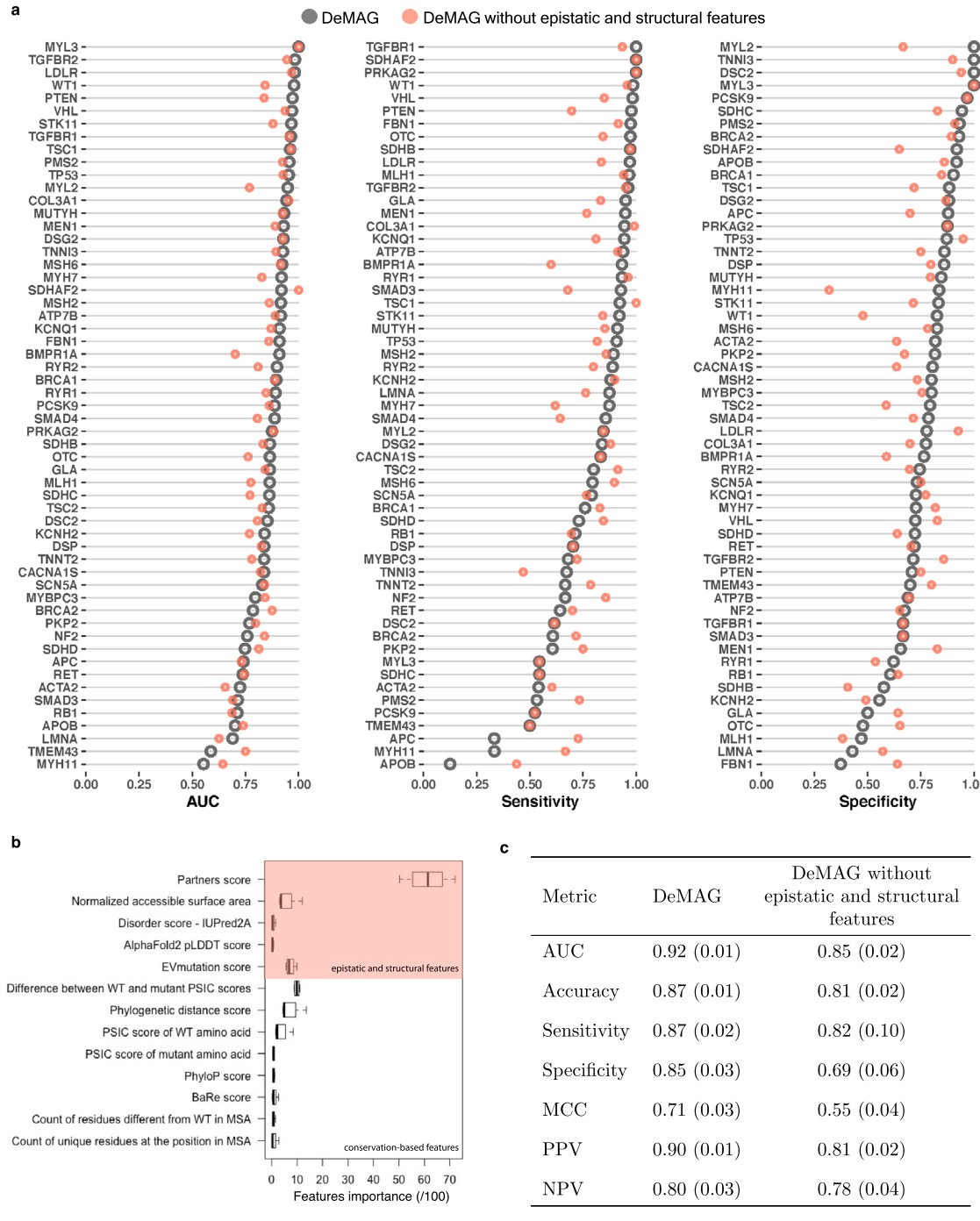

**Fig. 4 | Epistatic and structural features improve DeMAG performance, both within and across genes in the training set. a** Performance within genes: for most genes, variants are classified with high ROC-AUC > 70%. The high performance is also maintained for the classification of pathogenic (sensitivity >70%) and benign (specificity >70%) variants in most genes. **b** The feature importance of the 13 DeMAG features averaged across $n = 4$ cross validation (CV) folds (see Supplementary Table 2 for features description). The whiskers of the boxplots range correspond to ±1.5 times the IQR (inter quantile range). The lower (upper) bound of

the box of the boxplot corresponds to the 25th (75th) percentile. The center is the median. The partners score feature stands out from the others with a median importance of ~60%. **c** Performance metrics for DeMAG. The metrics represent mean values obtained by 4-fold CV with standard deviation in parentheses. Notably, DeMAG shows balanced performance with 87% sensitivity and 85% specificity as well as 90% positive predicted value and 80% negative predictive value. Removing epistatic and structural features provides consistently lower performance than the complete model.

Overall and at the single gene level, DeMAG has a balanced sensitivity and specificity (Fig. 4a, c), which corresponds to setting the threshold to 0.5 to interpret a variant as pathogenic.

## Epistatic and structural information improves performance

We investigated the contribution of each feature and observed that the partners score is the most informative one (Fig. 4b). In addition, a

structural feature, the normalized accessible surface area, is contributing at least as much as other conservation-based features, e.g., PSIC score[25]. In order to quantify the contribution of epistatic and structural features, we trained DeMAG without those features and observed a consistent decrease across all evaluation metrics (Fig. 4c). In particular, the specificity dropped from 85% to 69%, while the sensitivity decreased from 87% to 82%.

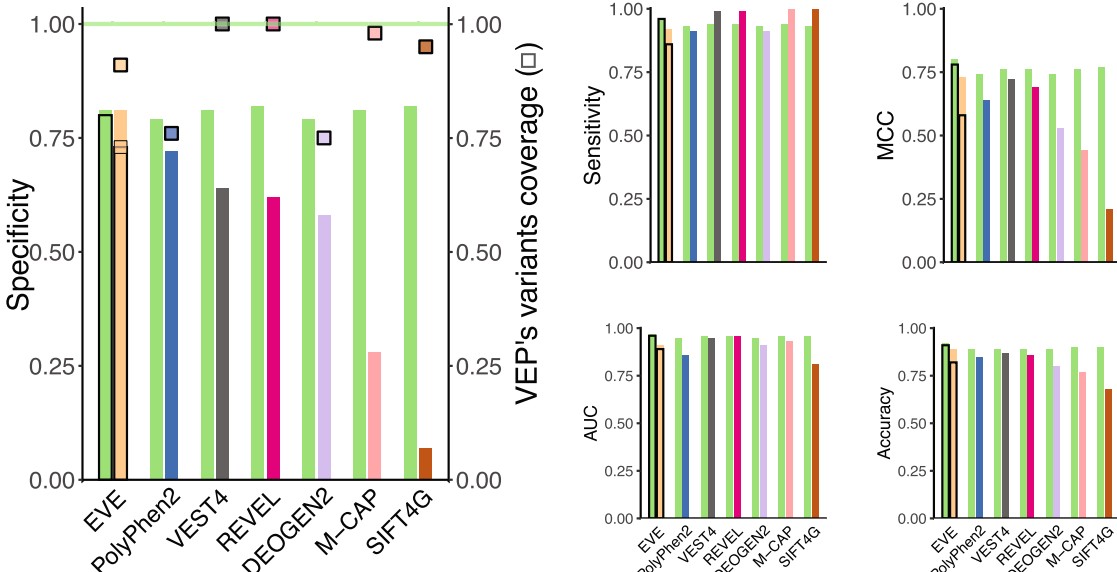

**Fig. 5 | DeMAG outperforms other VEPs on clinical variants.** Pairwise comparison of the performance of DeMAG and other VEPs on 1285 high-quality clinical variants from the ClinVar testing set (852 pathogenic and 433 benign variants), according to different performance metrics (specificity, sensitivity, Matthews Correlation Coefficient (MCC), ROC-AUC and accuracy). The left panel shows specificity and variants coverage of VEPs (squares) (e.g., SIFT has 95% coverage, see Table 1 for exact values). The green horizontal line ($y = 1$) indicates that DeMAG has 100% coverage, it has prediction for all variants. DeMAG's high specificity is on par with EVE on its 73% variants coverage (thin black bordersquare). For EVE there are 2 comparisons: the first (bars without black border) indicate the performance when uncertain class variants are excluded and the bars with black borders include the uncertain variants as well. We assigned their class as if EVE was a random classifier (Table 1). While all tools reach almost perfect ROC-AUC and sensitivity, DeMAG has the most balanced performance, namely the highest MCC (top right panel). For confidence intervals calculated on 1000 bootstrap samples refer to Supplementary Fig. 14.

## Table 1 | DeMAG outperforms other VEPs on clinical variants

| VEPs | Sensitivity | Specificity | Accuracy | MCC | AUC | Variants predicted (n = 1285) | VEP's coverage |
|---|---|---|---|---|---|---|---|
| DeMAG | 0.93 | 0.82 | 0.90 | 0.77 | 0.96 | – | – |
| SIFT4G | 1 | 0.07 | 0.68 | 0.21 | 0.81 | 1226 | 0.954 |
| DeMAG | 0.94 | 0.82 | 0.89 | 0.76 | 0.96 | – | – |
| REVEL | 0.99 | 0.62 | 0.86 | 0.69 | 0.96 | 1285 | 1 |
| DeMAG | 0.93 | 0.79 | 0.89 | 0.74 | 0.95 | – | – |
| DEOGEN2 | 0.91 | 0.58 | 0.80 | 0.53 | 0.91 | 964 | 0.750 |
| DeMAG | 0.93 | 0.79 | 0.89 | 0.74 | 0.95 | – | – |
| PolyPhen2 | 0.91 | 0.72 | 0.85 | 0.64 | 0.86 | 972 | 0.756 |
| DeMAG | 0.94 | 0.81 | 0.89 | 0.76 | 0.96 | – | – |
| VEST4 | 0.99 | 0.64 | 0.87 | 0.72 | 0.95 | 1280 | 0.996 |
| DeMAG | 0.94 | 0.81 | 0.90 | 0.76 | 0.96 | – | – |
| M-CAP | 1 | 0.28 | 0.77 | 0.44 | 0.93 | 1256 | 0.977 |
| DeMAG[a/b] | 0.96/0.96 | 0.81/0.80 | 0.92/0.91 | 0.80/0.78 | 0.97/0.96 | – | – |
| EVE[a/b] | 0.92/0.86 | 0.81/0.73 | 0.89/0.82 | 0.73/0.58 | 0.91/0.89 | 940/1165 | 0.731/0.906 |

Different performance metrics for DeMAG and seven popular variant effect predictors (VEPs). The test set is assembled from the ClinVar database, consisting of both pathogenic (n = 852) and benign variants (n = 433) submitted after the year 2017 (see Methods). The comparison in pairs guarantees that each predictor is evaluated on all the variants for which a prediction exists. DeMAG has 100% coverage and it is the most balanced across all the metrics. The comparison with EVE includes two values. The first value indicates the performance on the variants which are predicted by EVE as benign and pathogenic excluding the uncertain class (73% of all variants). The second value includes variants that EVE misclassifies as uncertain. We assigned them as if EVE was a random classifier.
[a]If EVE's Uncertain class variants are excluded from the testing set.
[b]If EVE's Uncertain class variants are assigned as if EVE was a random classifier.

We explored the contribution of epistatic and structural features at the gene level and observed an increase in sensitivity for genes with high proportions of pathogenic mutations (Supplementary Figs. 8a and 9). The specificity increased for genes with different proportions of pathogenic mutations, albeit mainly for genes with high proportions of benign mutations (Supplementary Figs. 8a and 9). The difference in

performance with and without epistatic and structural features appears to be independent of the number of training variants per gene (Supplementary Fig. 8b).

While it is evident that DeMAG's performance increased with epistatic and structural features overall, the improvement at the gene level is more challenging to assess.

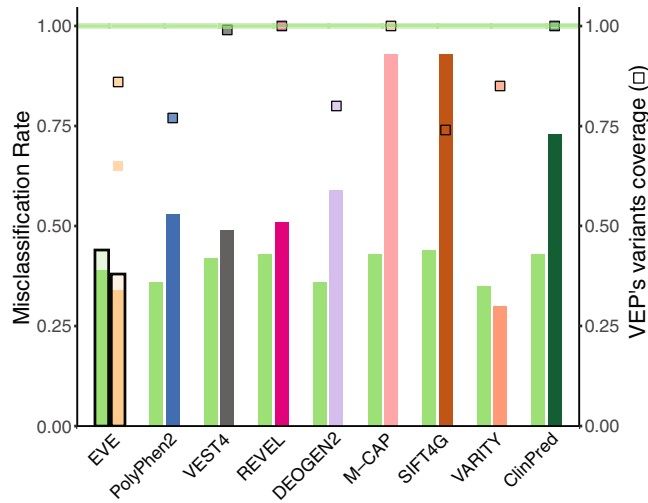

**Fig. 6 | Comparison of VEPs on population data.** Misclassification rate of benign common variants within the Estonian Biobank. The test set constitutes of 94 variants. On the y axis the misclassification rate and on the second y axis the VEP's variants coverage, identified by the colored squares. The green horizontal line ($y = 1$) shows that DeMAG has 100% coverage, as wells as REVEL, M-CAP and ClinPred. Among those tools DeMAG reaches the lowest misclassification rate. VARITY reaches the lowest misclassification rate but covers only 80% of variants. The comparison with EVE has 2 error bars: the first (black border and transparent color) indicates the performance when uncertain class variants are predicted as if EVE was a random classifier, the inner (and shorter) bar indicates EVE's performance excluding uncertain variants. While EVE has lower misclassification rate than DeMAG, it only gives certain predictions for 65% of variants. Although VARITY and EVE classify these 94 variants with the lowest misclassification rate, DeMAG has the lowest misclassification rate among VEPs with 100% predictions. For confidence intervals calculated on 1000 bootstrap samples refer to Supplementary Fig. 15.

## Many VEPs fail to predict functional effects observed in deep mutational scans

The concordance between VEPs and multiplexed assays of variant effects (MAVEs) on clinical data has been frequently reported[20,51–53], and both techniques are used to reevaluate VUSs according to ACMG-AMP guidelines[54]. Usually, the agreement between DMS data and VEPs is assessed through correlation coefficients and AUCs[55], which do not depend on a classification threshold that is needed for clinical decision-making.

We validated DeMAG as well as other VEPs against Deep Mutational Scanning (DMS) data as in prior studies[20,23,52], using data for 4 genes (*BRCA1*[53], *TP53*[56], *PTEN*[57], and *MSH2*[58]). Most variants in these assays are not yet annotated in ClinVar (51%), and 41% are ClinVar VUSs (Supplementary Fig. 4). Among the 7 VEPs evaluated, DeMAG performed best on *BRCA1* (35% Matthews Correlation Coefficient, MCC) and *PTEN* (27% MCC), while EVE performed better for *TP53* (39% MCC) and *MSH2* (38% MCC) (Supplementary Fig. 4).

The DMS data analysis indicates an overall poor performance of VEPs on functional data, at least on these four genes (Supplementary Fig. 4). Importantly, while sensitivity and ROC-AUC values are high (>80%) for all VEPs tested, the specificity values are worse than the one of a random classifier (Supplementary Fig. 4). This might be due to the high proportion of variants interpreted as functional by DMS data, e.g., for *MSH2*, 92% of single nucleotide missense substitutions are assessed as functional, and classified as pathogenic by VEPs. These results are in agreement with previous studies reporting weak performance of VEPs in predicting beneficial mutations (gain-of-function) in DMS data[52], and overall low concordance (Spearman correlation coefficient <50%) between VEPs and DMS data[18].

**Table 2 | Misclassification rate of common variants of the Estonian Biobank**

| VEPs | Misclassification rate | Variants predicted ($n = 94$) | VEPs coverage |
|---|---|---|---|
| DeMAG | 0.44 [0.32,0.56] | – | – |
| SIFT4G | 0.93 [0.86,0.99] | 70 | 0.74 |
| DeMAG | 0.43 [0.33,0.52] | – | – |
| REVEL | 0.51 [0.40,0.62] | 94 | 1 |
| DeMAG | 0.36 [0.25,0.46] | – | – |
| DEOGEN2 | 0.59 [0.47,0.69] | 75 | 0.8 |
| DeMAG | 0.36 [0.26,0.47] | – | – |
| PolyPhen2 | 0.53 [0.41,0.64] | 72 | 0.77 |
| DeMAG | 0.42 [0.32,0.52] | – | – |
| VEST4 | 0.49 [0.39,0.60] | 93 | 0.99 |
| DeMAG | 0.43 [0.33,0.52] | – | – |
| M-CAP | 0.93 [0.87,0.97] | 94 | 1 |
| DeMAG[a/b] | 0.39/0.44 [0.27,0.52]/[0.34,0.55] | – | – |
| EVE[a/b] | 0.34/0.38 [0.23,0.47]/[0.28,0.49] | 61/81 | 0.65/0.86 |
| DeMAG | 0.43 [0.33,0.52] | – | – |
| ClinPred | 0.73 [0.64,0.82] | 94 | 1 |
| DeMAG | 0.35 [0.25,0.45] | – | – |
| VARITY | 0.30 [0.20,0.40] | 80 | 0.85 |

The square brackets indicate the 95% CI calculated on 1000 bootstrap samples. The test set constitutes of 94 variants and 4 out of 9 VEPs have predictions for all variants (DeMAG, REVEL, M-CAP, ClinPred). Among those tools DeMAG reaches the lowest misclassification rate. Overall, VARITY reaches the lowest misclassification rate but covers only 80% of variants. The comparison with EVE has 2 rows: the first row indicates the performance when uncertain class variants are excluded; the second row indicates the performance that includes uncertain variants. Their class is assigned as if EVE was a random classifier. While EVE has lower misclassification rate than DeMAG, it only gives predictions for 65% of variants.
[a]If EVE's Uncertain class variants are excluded from the testing set.
[b]If EVE's Uncertain class variants are assigned as if EVE was a random classifier.

## DeMAG outperforms existing tools on clinical data

As VEPs often collect variants from similar database sources, it is essential to benchmark different predictors against an unbiased testing set to avoid type 1 circularity[21]. To compare our performance with commonly used VEPs (PolyPhen-2[8], SIFT4G[59], REVEL[11], DEOGEN2[60], M-CAP[10], VEST4[9], and EVE[18]), we designed a clinical testing set comprising high-quality variants submitted to ClinVar after 2017. We assume that, as the most recent supervised method we benchmarked with was published in 2016, none of those methods used these variants for training as they were not yet part of the ClinVar database. We retrained DeMAG without these held-aside variants for testing, and used this model to make predictions on the ClinVar variants (Supplementary Table 3 and Supplementary Fig. 10). The testing set had 852 (66%) pathogenic and 433 (34%) benign variants. As not all VEPs have predictions for these variants, we benchmarked DeMAG in pairs (Table 1 and Fig. 5a). Overall, DeMAG reached the highest specificity, accuracy, and MCC values (Table 1 and Fig. 5). DeMAG's performance is consistently high across the different evaluation metrics, while other VEPs present lower specificity compared to sensitivity (e.g., DeMAG 82% vs. 93%, and REVEL 62% vs. 99%).

It should be noted that EVE's high accuracy (89%) dropped to 82% when we included variants that the authors assigned as uncertain, although they are actually annotated as benign or pathogenic variants in ClinVar (Table 1 and Methods section). In addition, only DeMAG and REVEL predict 100% of variants, while EVE has the lowest coverage (73%) (Table 1 and Fig. 5). While each tool has different strengths, DeMAG outperforms all other methods tested in at least one evaluation metric reported.

**Table 3 | ClinVar clinical significance of common variants of the Estonian Biobank**

| Clinical significance | Review stars | | | |
|---|---|---|---|---|
| | 0 | 1 | 2 | 3 |
| Benign | 0 | 3 | 99 | 39 |
| Pathogenic | 1 | 1 | 2 | 0 |
| VUS | 1 | 48 | 106 | 1 |
| Conflicting pathogenicity | 0 | 168 | 0 | 0 |
| Not annotated in ClinVar | 117 | | | |

The table shows the ClinVar clinical significance of the common variants from the Estonian Biobank according to the review status stars which increase with the confidence of a variant assessment. For example, among high-quality (review status with at least 2 stars) annotations most variants are benign and VUSs.

We also benchmarked against common variants in the Estonian Biobank[61]. While variants from the Estonian Biobank are not yet part of gnomAD, 80% were already annotated in ClinVar (Table 3). Most variants were VUSs (33%) and high-quality benign variants (30%). We evaluated variants, not already annotated in ClinVar or in our training set, and we filtered the variants based on MAF greater than the corresponding disease prevalence (Supplementary Data 2), resulting in a total of 94 benign variants. For this analysis we also compared DeMAG with some of the most recent supervised VEPs, VARITY[12], and ClinPred[62]. Since those tools were trained on ClinVar variants of our test set, we did not include them in the previous ClinVar analysis. VARITY has the lowest misclassification rate (30%) followed by EVE (34%) and DeMAG (39%), however, both VARITY and EVE predict only 85% and 65% of variants, respectively (Table 2 and Fig. 6). DeMAG has the lowest misclassification rate (43%) among VEPs (REVEL 51%, ClinPred 72% and M-CAP 93%) that predict 100% of variants (Table 2 and Fig. 6).

DeMAG was designed and trained to aid clinical variant interpretation of a specific, actionable gene set. However, the model and the partners score framework can be extended to other genes that have enough variants with known phenotypic information. Therefore, we evaluated the generalizability of DeMAG on other clinically relevant genes. We only included genes with at least 5 benign and 5 pathogenic high-quality variants (review status of at least 2 stars) in ClinVar (Supplementary Data 1). We used the same model trained on the 59 ACMG SF genes and evaluated its performance on the 257 new genes. The results demonstrate that DeMAG generalizes well, reaching 91% sensitivity and 85% specificity (Supplementary Table 4). We anticipate that with the growth of clinical data, DeMAG will be used to evaluate variants in even more genes by taking advantage of the partners score framework.

## Discussion

As genomic sequencing becomes more commonplace in clinical practice and research, the interpretation of missense variants remains a major challenge. Correctly classifying the pathogenicity of variants is essential to translating genomic information from actionable genes into clinical care. We developed DeMAG, a specialized VEP that reaches high performance for such actionable disease genes (ACMG SF) (Fig. 4a, c). It demonstrates superior balance between sensitivity and specificity (Fig. 5 and Table 1), and which has utility for variant prioritization, rare variant studies, and to reclassify VUSs in the ACMG SF genes. As many as 16% of missense VUSs in ClinVar are within the 59 ACMG SF genes, underlying the value of a specialized classifier. While exome-wide predictors have a wide range of uses in basic research, here we show that a specialized classifier reaches higher performance on clinically actionable genes, and should be prioritized in translational research (Table 1 and Fig. 5).

The assembly of a high-confidence balanced training set is crucial for the development of supervised predictors. For example, the

ClinVar *Review status* provides a system to evaluate the review quality and agreement related to confidence of a variant assessment. Thus, we included only variants whose clinical interpretation is shared among different submitters, i.e., 2 or more review status stars. The joint analysis of clinical annotations between databases, namely ClinVar and HGMD, allowed the removal of potentially conflicting or lower-quality variants. Indeed, we removed almost 40% of disease-causing variants in HGMD that were interpreted in ClinVar as VUSs (Supplementary Fig. 5), and included a large number of new putatively neutral variants which are statistically unlikely to be highly penetrant disease variants (Supplementary Fig. 2a and b). As clinical databases become increasingly important repositories for genetic variation in relation to human health and disease phenotypes, it is crucial to implement quality control pipelines to include only variants with non-conflicting and clear interpretations.

As many VUSs are identified in diagnostic testing, many studies are focusing on VUS assessment and reclassification[54,63]. For instance, Dines et al.[63] reclassified *BRCA1* exon 11 as a cold spot, suggesting a benign reinterpretation of variants located within that region. DeMAG predictions for that region agree with such reclassification (Supplementary Fig. 11). On the other hand, the reassignment of the *BRCA1* coiled-coil domain (1393–1424) as a moderate benign region is in disagreement with previous study that showed that mutations in that region disrupt the complex formation with *PALB2*, which would impair the Homologous Repair (HR) mechanism[64]. DeMAG agrees with this work and it classifies at least 45% of all possible missense substitutions in that region as pathogenic (Supplementary Fig. 11).

In addition to AUC-ROCs, VEPs should include other performance characteristics and metrics, especially when training on unbalanced data[65]. We have reported several performance metrics when benchmarking DeMAG (Tables 1 and 2 and Figs. 5 and 6) that confirmed how several popular VEPs often fail to correctly classify benign variants[23] (Tables 1 and 2 and Figs. 5 and 6). As computational evidence is commonly used and one of the classification criteria, bias in overestimating pathogenicity contributes to labeling more variants as pathogenic in publicly available databases.

The epistatic and structurally derived features are informative, as DeMAG has inferior performance without these features for all metrics considered (Fig. 4c). Despite the overall improvement, there are a few genes that do not benefit from those new features (Fig. 4a and Supplementary Figs. 8 and 9). This might be due to the imbalanced nature of pathogenic and benign training variants within these genes (Supplementary Fig. 2c). The performance of genes that harbor almost only pathogenic (or benign) mutations will be dominated by high sensitivity (or specificity), so an improvement in the dominant metric will result in a substantial drop of the other one. For instance, the new features increase sensitivity in *FBN1*, but as the gene has 93% pathogenic mutations, the specificity drops by 27% (Supplementary Fig. 9 and Fig. 4a). The same happens for *APOB*, *MYH11* and *APC*. These genes harbor benign variants in proportions >86%, and indeed, an increase in specificity corresponds to more than 30% drop in sensitivity. Though we are able to improve the overall balance in performance characteristics, some clinically actionable disease genes have significant biases which pose challenges for variant interpretation.

We observe that evolutionary coupled positions and spatially proximal ones are enriched for the same phenotypic effects, and might serve to identify functional sites, e.g., domains (Supplementary Table 7). In agreement with previous reports[12,16], we confirm that the traditional conservation paradigm to interpret human coding missense mutations should be complemented with epistatic and structural information.

Although DeMAG was trained on the 59 ACMG SF genes, it generalizes well to an extended set of 257 ClinVar genes (Supplementary Table 4). Extending to additional genes would require abundant clinical data, which is not yet available for most genes (Supplementary

Fig. 12 and Supplementary Data 1). As large population sequencing datasets become available, supervised predictors like DeMAG may be able to further improve assessment of variant effects.

In conclusion, we anticipate that our tool and the web server (demag.org) will facilitate variant assessment and clinical decision-making. Moreover, the newly developed features can be applied to other genotype-phenotype predictors and be generalized to other genes and organisms.

## Methods

### Training dataset

Variant Call Format (VCF) files were collected from clinical and population databases. We downloaded the ClinVar[4] VCF file, version 2021.05 and retained variants with a review status of at least 2 stars, with either a 'pathogenic' or 'benign' clinical significance (including likely benign and likely pathogenic labels). Variants of conflicting interpretations were excluded as well as variants which had only somatic labels. We used the Human Mutation Gene Database[40] (HGMD), version 2020.03, to extract additional pathogenic mutations. We filtered for disease mutations (DMs) and retained variants that were not already annotated in ClinVar (Supplementary Fig. 5). With this filtering we removed HGMD variants with a VUS label in ClinVar (26%), as well as low quality (zero and one review status star) ClinVar pathogenic variants (27%) and ClinVar benign variants (1%). The PolyPhen-2 HumVar training dataset derived from UniProtKB release 2021.01[66] was used to collect both pathogenic and benign variants.

We added common variants to the benign set from the Genome Aggregation Database[41] (gnomAD), the NCBI ALFA[67] (Allele Frequency Aggregator) project release 20201124, country-specific sequencing projects, i.e., Korea[42] (KRGDB) and Japan[43] (3.5KJPNv2,), and variants from human orthologues (PrimateAI[30] and Human/Chimpansee substitutions extracted from the UCSC Genome Browser Multiz alignments of 100 vertebrates). Non-human primate variants were collected from the primateAI database but only *Chimpansee* and *Bonobo* species were considered as the most closely related apes to humans. We treated non-human primate polymorphisms as benign variants. In addition, we considered variants from human population data as benign, if their minor allele frequency (MAF) was greater than the disease prevalence (BS1 classification of ACMG guidelines[13,14]). Disease prevalence values were collected from Orphanet and MedlinePlus (https://www.orpha.net, https://medlineplus.gov/). In case of multiple disease prevalence values associated to a phenotype we chose the highest value, according to a conservative strategy, and when unavailable, a MAF filter >0.5% was applied (Supplementary Data 2). These variants collectively were considered benign to train and test the model.

Both duplicates and conflicting variants, i.e., variants reported both as pathogenic and benign among different sources, were removed from the final training set. The number of training variants among different sources and different genes are shown in Supplementary Fig. 2.

### Clinical testing dataset

The primary clinical testing set (852 pathogenic and 433 benign variants) was built from the ClinVar database. To ensure the independence of the testing set, we only considered variants submitted to ClinVar after December 2017 (Supplementary Fig. 10). Because the newest supervised method we benchmarked with was published in 2016, these variants were not used in the training pipeline of any of the predictors. As we used ClinVar variants for training, we trained a different model for the performance testing (Supplementary Table 3), excluding the variants held aside for testing (Supplementary Fig. 10). This ensured unbiased comparison of DeMAG to other predictors.

### Functional variants testing set

In order to investigate the concordance between DeMAG predictions and experimental data, we used DMS data for *BRCA1*, *TP53*, *PTEN*, and *MSH2*. All datasets were collected from the Supplementary materials of the respective papers[53,56–58]. When possible, we assessed the concordance between different experimental replicates to ensure a robust functional score for each variant. For *BRCA1*, two scores were available and as the correlation and the variance explained was 81% and 65% respectively, we included all the variants. The authors assessed variants' functional scores in three categories: loss of function (LOF), intermediate (INT), and functional (FUN). We did not evaluate the intermediate class. After removing overlapping variants with our training set, we evaluated 1587 variants: 1268 (80%) FUNC and 319 (20%) LOF. For *PTEN*, 8 different scores were available. Since the correlation pattern among the replica was variable, we only evaluated variants whose standard deviation among all available 8 scores was smaller than 10%. In this case as well, we did not evaluate uncertain functional categories, namely *possibly wt-like* and *possibly low*. We evaluated 34 FUNC (64%) and 19 LOF (36%) variants. For *MSH2*, we could not analyze the concordance among different replicas, as only one score was provided. The total number of variants analyzed was 5075: 4737 (93%) FUNC and 338 (7%) LOF variants. The last gene we analyzed was *TP53*, for which we did not have more than one functional score but agreement between replica was already assessed by the authors. We evaluated 1017 variants: 714 (70%) FUNC and 303 (30%) LOF variants.

### Common variants testing set

We assembled another testing set of putatively benign variants from the Estonian Biobank[61]. To define benign variants, we applied the same rule as for the common variants in the training set and used MAF threshold greater than the disease prevalence (Supplementary Data 2). In order to design an independent testing set, we removed variants that were present in our training set as well as variants with a ClinVar interpretation. As those variants were still not part of gnomAD at the time we wrote the manuscript, it is unlikely that any VEPs used them in their training sets. The test set consisted of 94 variants. We calculated the misclassification rate and we constructed its 95% confidence interval (CI) from 1000 bootstrap samples, using the 25th and 975th value of the misclassification rate resampling distribution.

### Pathogenicity scores

Pathogenicity scores were collected through dbNSFP[68] v4.1a command-line application. We have downloaded scores for SIFT4G v2.4, VEST v4.0, PolyPhen-2 v2.2.3, M-CAP v1.3, DEOGEN2, and REVEL. To calculate the accuracy, we used the threshold as recommended by the authors, which is 0.5 for all the methods except for M-CAP which is 0.025 and SIFT4G which is 0.05. For EVE, we downloaded the predictions from the web server (https://evemodel.org/download/bulk) on 2021.12.16 and used the columns "EVE_scores_ASM" and "EVE_classes_75_pct_retained_ASM" (as reported in the web server) respectively for the continuous probability score and categorical classification. EVE does not provide a unique threshold, rather a gene-based predefined categorical feature with three different levels: pathogenic, benign, and uncertain. When uncertain variants corresponded to misclassified variants i.e., ground-truth pathogenic and benign variants, we assigned them as if EVE was a random classifier. VARITY and ClinPred precomputed scores were downloaded respectively on 2022.01.05 and on 2022.11.15 from their web servers.

### Variant annotation

We used MapSNPs from PolyPhen-2 v2.2.3 (http://genetics.bwh.harvard.edu/pph2/dokuwiki/downloads) annotation tool to map the genome assembly hg19/GRCh37 variants coordinates to missense

coding SNPs. Only variants mapping to known canonical transcripts according to the UCSC Genome Browser were retained.

## Sequence-based features

We used the PolyPhen-2 v2.2.3 pipeline to annotate DeMAG features (http://genetics.bwh.harvard.edu/pph2/dokuwiki/downloads). A complete list and description is available at the PolyPhen-2 v2.2.3 Wiki page (http://genetics.bwh.harvard.edu/wiki/pph2/appendix_a). The new features are annotated separately (see sections below). IUpred2A scores were collected using the command line tool[69] (https://iupred2a.elte.hu/).

## Epistatic and structure-based features

**EVmutation**. EVmutation scores[17] were obtained using the EVcouplings Python package, version v0.1.1[70]. EVmutation scores were computed for protein residues covered by the multiple sequence alignment (MSA) of the corresponding protein sequence. In order to maximize the alignments coverage, and to cover regions other than the most conserved domains, we optimised alignments. In particular, protein sequences were tiled in regions of 100 residues with overlapping windows of 50 residues, i.e., 1–100, 50–150. The MSA was then computed for each tiled region and for five different bit score thresholds (0.1, 0.2, 0.3, 0.4, 0.5). For each of these combinations we calculated the number of sequences in the MSA and the skewness of the Evolutionary Couplings (EC) distribution. We merged adjacent regions if either the number of sequences in the alignment was >5 times the length of the region or if the skewness of the EC distribution was >1. At each step, we repeated the align and couplings stages of the EVcouplings framework. We repeated these steps until no more adjacent regions could be joined together. Then, we computed the mutation stage to obtain the EVmutation score. The final alignment coverage for the ACMG SF genes is shown in Supplementary Fig. 13a. While the pre-computed EVmutation scores available on the webserver (downloaded on 2022.09.22) cover only 37% of ACMF SF v2.0 sites, with our MSA pipeline we increased the coverage to 64% of residues.

**Partners score**. The partners score is derived from co-evolutionary partners, i.e., evolutionary coupled residue positions, and structural partners, i.e., spatially close residue positions (Fig. 2). Co-evolving or coupled residue positions were identified using the EVcouplings framework. First, we obtained a list of coupled residue pairs, and then we annotated the label (phenotypic effect) as in our training set for each residue. If a residue position was associated with both pathogenic and benign variants we assigned the label "mixed". We excluded residue positions that were coupled to only not annotated residues in the training set. A residue that is not annotated is a residue that does not have any variants with known label in our training set. We assigned a score to each residue: 1 for pathogenic positions, −1 for benign and 0 for mixed or not annotated ones. The residue score is the sum of the scores of all co-evolving positions (Fig. 2b) of any given residue position. Then, we trained a mixture discriminant analysis model on the residue score distribution of the training set variants. First, the model estimates the density of the residue score distribution for the pathogenic and benign variants independently. Next, the model predicts for each residue position the posterior probability of belonging to the benign and pathogenic class given the residue score and the prior probability of being a pathogenic or benign position as in the training set. The partners score of co-evolving residue positions is the posterior probability of belonging to the pathogenic class (Fig. 2d).

The significance of coupled residues is determined by their location in the EC score distribution. A probability model has been defined to identify strong coupled positions[70]. The higher the probability the more likely the residues co-evolve. In order to select the best probability threshold, we trained the mixture discriminant analysis model for different cutoffs using cross-validation. We selected the probability

cutoff (0.6) which gives the smallest difference between sensitivity and specificity (Supplementary Fig. 6b).

Similar approach was used for spatially close residue positions as in AlphaFold2 3D models. In order to select the Ångström distance threshold for considering a pair of residues as contacting in 3D space we trained different models with different cutoffs (4-11 Å). We selected 11 Å as the best distance, which gave the smallest difference between sensitivity and specificity (Supplementary Fig. 6b). We did not consider larger distances to avoid introducing protein-specific properties rather than residues-based ones.

The residue pairs that co-evolve and are contacting in 3D space overlap and as a consequence the partners scores based on pairs defined by co-evolution and 3D structure correlate (Supplementary Fig. 6c). Therefore, we combined them and took the union of all scores to increase the coverage of positions. In case of overlap, when both scores were available for a position, we chose the score based on the spatially close residue pairs.

The mixture discriminant analysis approach was implemented using the mclust[71,72] package in R. The best model is internally selected by Bayesian Information Criteria[73] (BIC) and it has 3 Gaussian components with variable variance for the density of the residue score for pathogenic variants and 4 gaussian components with equal variance for the benign ones. Given the density estimation of the residue score and the prior probabilities (i.e., frequency) of the benign and pathogenic variants, the mixture model predicts the posterior probability of belonging to both classes (pathogenic and benign). We define the partners score to be equal to the posterior probability of pathogenicity (Fig. 2d). In total, 49,822 residues have a partners score.

## 3D models

We collected protein 3D models built by AlphaFold2 model (version v2.1.0, 2021.08.12) that resulted in 100% residue coverage among the ACMG SF genes (Supplementary Fig. 3). For long genes (*APC, APOB, BRCA2, DSP, FBN1, RYR1, RYR2*) AlphaFold2 produces different overlapping models that we combined to obtain one single complete model. The models are ~1400 aa long with non-overlapping regions of ~200 aa that cover the full sequence.

## Cross-validation scheme

In order to select features, hyperparameters and the best probability and Ångström distance cutoffs, we trained the models using a cross validation scheme. The cross-validation scheme ensured that each testing fold contained different proteins than the ones in the training folds. This prevents bias due to training and testing within the same protein. In addition, each testing fold should have a distribution of the pathogenic and benign class that reflects the one of the training set. To respect these two principles, we considered 4 CV-folds for model training and hyperparameters selection and 5 CV-folds for the features selection pipeline.

## Feature selection

Training set variants were annotated with a total of 91 features. Most features were annotated with PolyPhen-2 v2.2.3 pipeline and a description of each feature can be found here (http://genetics.bwh.harvard.edu/wiki/pph2/appendix_a). After removing gene-based features, 39 features were retained. In order to select the most discriminative features to train the model with, we trained a univariate logistic regression model. The CV strategy is explained above (see Cross-validation scheme subsection). The features that had an ROC-AUC above 0.7, while ensuring a corresponding sensitivity and specificity >0.5, were selected. The final model was trained with a total of 13 features. The feature selection process was repeated twice: once for DeMAG and second time for DeMAG without the ClinVar testing set. The same set of features were selected for both models, see Supplementary Tables 1a and 2b.

## Machine learning models

DeMAG was trained with a gradient-boosting model with classification tree as base learner and Bernoulli deviance as loss function. R package "gbm" version 2.1.8 was used for training[74]. We trained the model with the 13 features selected during the feature selection pipeline. We implemented a grid search for two of the parameters of the gbm function: shrinkage and interaction depth. The combinations evaluated were 9, resulting from 3 values for the shrinkage parameter (0.001, 0.0055, 0.01) and for the interaction depth (1, 2, 3). The best combination of parameters was selected based on performance in 4-fold CV: the models were ranked based on the smallest difference between sensitivity and specificity and if more than one model satisfied the condition the model with the highest sensitivity and specificity was selected (Supplementary Tables 1b and 2). As for the feature selection, the grid search was performed for DeMAG without the ClinVar test set as well. Once we identified the best parameters, the gbm model was trained with 4-fold CV to inspect the robustness of the 4 models' performance and to investigate any potential biases in the training set. The final model was then trained on the complete dataset. The gradient-boosting model was chosen over other ensemble machine learning techniques such as Random Forest because it explicitly handles missing values (6% of features annotations in the training set), namely for each decision in the tree there are not only the left and right nodes but a missing node as well. Missing information is thus treated as information, in the sense that rather than attributing an imputed value a priori, the model's algorithm considers the missing value as another value of the specific predictor and as such is included as another node.

## Testing the generalizability of DeMAG on an extended set of 257 genes

We evaluated DeMAG on an extended set of 257 ClinVar genes. We used ClinVar VCF file version 2022.08.12. We retained only genes with at least 5 benign and 5 pathogenic high-quality (at least 2 review status stars) variants and we excluded genes longer than 1800 as AlphaFold2 models are less reliable for long genes. See above subsections for features annotation. The only difference in features annotation was that we used pre-computed EVmutation score (https://marks.hms.harvard.edu/evmutation/downloads.html) and that we designed the partners score based only on structural partnerships (spatially close residues according to AlphaFold2 3D models (version v2.3.0, downloaded on 2022.09.06)).

## Data analysis and visualization

All statistical analysis was done in R (see the github repository for the lists of packages used). The figures and tables were made with R, Adobe Illustrator, Biorender.com and LaTeX.

To visualize protein 3D models we used Pymol. The webserver was created with RShiny app.

## Reporting summary

Further information on research design is available in the Nature Portfolio Reporting Summary linked to this article.

## Data availability

The training set (with the exception of the HGMD variants) and all the test sets generated for this study are available on gitlab (https://git.mpi-cbg.de/tothpetroczylab/DeMAG) and on DeMAG webserver (https://demag.org/) and have been deposited at this repository: https://doi.org/21.11101/0000-0007-FB84-9. HGMD data were available to the authors under a subscription data use agreement which prohibits sharing variant data from HGMD Professional (QIAGEN). Users and developers may not make HGMD data publicly available. (https://www.hgmd.cf.ac.uk/docs/disclaimer.html). The variants from the Estonian Biobank have been obtained through the process described here: https://genomics.ut.ee/en/content/estonian-biobank. The collection of the raw data is described in the Supplementary Table 5.

## Code availability

The code is available on gitlab (https://git.mpi-cbg.de/tothpetroczylab/DeMAG) and the webserver at https://demag.org/.

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

## Acknowledgements

This work was supported by NIH/NHGRI grant (R01-HG010372) and by Max Planck Society MPRGL funding. We thank the Institute of Genomics of the University of Tartu for sharing genotype data from 4776 individuals. We especially thank Prof. Dr. Andres Metspalu and Krista-Roberta Saviauk for the Estonian Biobank data. We thank Shamil Sunyaev for many scientific discussions and for critical reading of the manuscript. We further thank Cedric Landerer for helping with the manuscript during the revision process, Hanna Josephine Wiederanders for the initial contribution to the partners score, Michele Marass and Jonas Pöhls for providing valuable feedback on the manuscript, HongKee Moon for the technical support, and the MPI-CBG Computer Services and Scientific Computing Facility for their support.

## Author contributions

F.L. contributed to the development of the method, collected the data, implemented the code, designed, and ran the data analysis, created figures and tables and drafted the manuscript. I.A.A. contributed to the development of the method and to the implementation of the code, and designed and run data analysis. C.A.C. contributed to the development of the method, collected data, and supervised the project. A.T-P. contributed to the development of the method, designed data analysis, and envisioned and supervised the project. All authors provided regular feedback to all aspects of the work and contributed to the writing of the final manuscript.

## Funding

## Competing interests

The authors declare no competing interests.
