## [Peer Review File · Nature Communications]

DeMAG predicts the effects of variants in clinically actionable genes by integrating structural and evolutionary epistatic featuresREVIEWER COMMENTS

Reviewer #1 (Remarks to the Author):

DeMAG is a supervised variant effect predictor for assessing the likely effect of missense variants in 59 disease-related human genes.

The main novel feature of DeMAG that sets it apart from the large number of competing tools in this space is the development of a "partners score", based on the evolutionary interdependence of variants within close proximity within protein structures. This work in particular has been made possible by the recent release of high quality human protein structure predictions by AlphaFold2.

In addition to the described method, the paper is accompanied with an interactive searchable website that provides the resulting predictions on variants in the 59 selected genes.

As with all variant effect predictors in this space, the main challenge is developing a classifier is the identification of sufficiently large and reliable truth sets. In this situation, the main source of "truth" is to be found in curated clinical databases, predominantly ClinVar. Naturally, this greatly limits the amount of high quality training data available, and the authors of this paper have gone to considerable lengths to develop a balanced training data set (and also sufficiently large and independent testing sets).

The performance of DeMAG is compared with several other popular variant effect prediction tools. The author's note that, in testing, many of the assessed tools have comparably good sensitivity, however, except for EVE, most other tools have relatively poor specificity when compared to DeMAG.

The authors also demonstrate the value of the "partners score" in overall classification performance of DeMAG by comparing to a version with the "partners score" removed, with specificity particularly affected.

While the method is novel, interesting and is demonstrated to have good predictive performance compared to a selection of popular competitors, I have a number of reservations about the manuscript as it is currently written. I will detail these issues below. I believe that these issues would need to be addressed if the paper was eventually to be accepted for publication.

The most significant limitation of DeMAG is that it currently only applies to 59 genes, thus has limited apparent generalisability. The other predictors in the comparison typically can be applied to thousands of human genes. Obviously there is a tradeoff to be made here. It is clear that better predictors can be made when they are designed to work on a smaller set of genes. In the extreme limit, a carefully chosen single-gene predictor should have the best performance of all. Extending a classification algorithm to a wider domain of input cases is usually going to come at a loss of prediction performance. This is especially important when comparing the results of DeMAG to its nearest competitor EVE. For most test metrics, except perhaps coverage, EVE performs very well compared to DeMAG. However, and most importantly, EVE's publication covers 3219 human genes; a couple of orders of magnitude more than DeMAG. In fact the authors of the DeMAG paper seem reticent to mention this fact, as it is not until deep into the paper that the number of covered genes is mentioned. This is a serious omission. I would expect the number of supported genes to be mentioned in the Abstract and the Introduction of the paper.

Given that the performance of EVE is quite comparable to DeMAG, and that EVE is applicable to a much wider number of human genes, I think the authors must provide a deeper comparison of the two methods, and a stronger argument as to why the "partners score" approach adds value above the method employed in EVE. One is left wondering whether it would be better to add the partners score to EVE, rather than writing a whole new classifier.

While it is pleasing to see that the manuscript is accompanied by an interactive searchable website (<https://demag.org/>) I could not find any way to download the entire DeMAG classifications in one bulk file. Many other classifiers make this information available in bulk. Searching for individual variants is somewhat useful, but it is hard to incorporate this data into other analyses or perform comparisons with other methods without a bulk download. It will not be a large amount of data, so I do not think there is any technical reason why a bulk download is not provided. If such an option is available, I was not able to find it in the interface of the website, and therefore I suggest that such a feature be made prominent.

Overall the paper was relatively easy to understand, however, there were many parts where I felt that the writing was vague, unclear or awkwardly phrased. It seemed as though the writing was rushed, and it would benefit from further proof reading. This was particularly apparent in the Supplementary methods section. On this topic I have provided a non-exhaustive list of parts of the paper that would benefit from redrafting.

I commend the authors for making their software available as well. This does aid to the reproducibility of the work. However, I would say that I would struggle to reproduce the method from the description in the paper and the Supplementary materials, and I have previous experience implementing variant effect prediction tools.

Specific comments about the text of the manuscript are below:

Human gene names should be italicised.

(page 3) This sentence needs clarification: "Indeed, human pathogenic variations appear as neutral substitutions in closely-related orthologous from other species." Surely this is not true for all human pathogenic variations. Is there a citation for this? Also, should it be "orthologs"?

(page 3) "in clinically actionable disease genes in the ACMG SF v2.0 list" you should say how many genes there are in this list. Also this piece of information should be mentioned in the abstract. It is an important detail.

(page 3) "Under and overdiagnosis as well as patients' psychological burden due to lack of evidence to support variants' pathogenicity may result in increased costs for the healthcare system." This seems quite speculative. It might be true, but is there any data to back up this claim?

(page 5) "Many existing VEPs do not explicitly attempt to balance sensitivity and specificity." This is a bold far reaching claim that must be supported by evidence. Also the word "many" is vague and imprecise. It reads more like an opinion or conjecture than a fact.

(page 6) "While EVE and DeMAG outperform other tools, their misclassification rate is still high on DMS datasets, which underlies a potential limitation for the clinical application of such data." Is the implication that that DMS data is misleading or the predictions are wrong?

(page 6) This sentence is confusing, please clarify: "As the most recent supervised method was published in 2016, none of the VEPs were trained on those newer variants."

(page 8) This comes out of the blue with no mention previously in the paper: "Nevertheless, the comparison between AlphaFold2 and IUPred2A showed how high-quality predictions are still missing for many ordered regions (Supplementary Fig. 6), highlighting how the mystery of protein folding is yet to be understood." This needs some kind of previous description.

(page 10) "experimental replicas" replicates?

(page 11) python -> Python

(page 12) rephrase: "in a connection with only not annotated residues"

(page 12) "residues-based" -> residue based

(page 12) this is unclear and imprecise: "thus we combined them and in case of overlap we took the spatially close residues score"

(page 12) "The best model selected by BIC" I believe this is the first mention of Bayesian Information Criteria (BIC). This probably deserves a bit more explanation, not all readers will be statisticians or data scientists.

(page 12) suggest rephrasing "We considered the posterior probability of pathogenicity as the partners score" to something like "We define the partners score to be equal to the posterior probability of pathogenicity"

(page 12) "Eventually" avoid this word in this context. It doesn't suit the scientific description of a method.

(page 12) rephrase "by dividing the times the t-statistic"

(page 13) "Since July 2021 we updated our 3D data with AlphaFold2 models" this part of the method is very unclear. If AlphaFold2 models have made substantial changes to the method from previous iterations, then why mention anything before this? Does this not obviate the need to discuss the previous methods?

(page 20) X axis labels for the comparisons would really help, otherwise reader has to compare colours.

Reviewer #2 (Remarks to the Author):

Overall, this was an interesting study. I particularly liked the idea of the novel 'partners score' feature and its predictive ability seemed to be excellent. The paper is well presented, and I have little doubt as to the performance of DeMAG in the 59 clinically actionable genes. My concerns primarily regard the scope of the study, the other VEPs benchmarked against and the availability of the data.

Major concerns

Study scope: It is mentioned in the discussion that up to 30% of VUS in ClinVar are from ACMG genes. This means that DeMAG cannot make predictions on $\geq 70\%$ of identified VUS. I find it unlikely that many non-specialist variant annotation pipelines (many of which still use SIFT and PolyPhen-2) would add a method which is unable to make predictions in a majority of variants.

DeMAG is only compared to supervised methods from pre-2016 (plus EVE) to counter the issue of type-1 circularity. Unfortunately this leaves us uncertain about its performance levels against more state-of-the-art predictors (e.g. VARIETY or ClinPred). The claim in the abstract that DeMAG out-performs existing VEPs has only been shown to be true for VEPs 8+ years old. As a suggestion, the DMS analysis could be expanded to include some more modern VEPs.

Features for the model other than the partners score were primarily taken from the PolyPhen-2 feature pipeline, many VEPs have used other innovative useful features since PolyPhen-2, it would have been interesting to see if any key features from other VEPs improved performance.

The amount of hyperparameter optimisation performed seems rather small (3-value grid search across 2 parameters), given the amount of effort put into creating and evaluating the partners score feature, this lack of optimisation for the final model was surprising.

A lot of care was used to create manually curated multiple sequence alignments to generate the evolutionary couplings that form the basis of the partners score. If EVE or SIFT (or even EVmutation) were run with these high-quality alignments rather than their native ones, perhaps they would perform better?

There is nowhere to download the full set of predictions! This stumps external benchmarking efforts.

Minor concerns

Citations for “The uncertainty about the pathogenicity of a variant may pose a psychological burden for patients” (10.1093/annonc/mdt312)

“Unsupervised methods, such as DeepSequence¹², EVmutation¹³ and EVE¹⁴ are agnostic to variant labels as they infer functional effects from multiple sequence alignment (MSA) alone”

Not all unsupervised methods infer effects from a MSA, for example Eigen.

Putatively benign variants is never defined.

Human Mutation Gene Database -> Human Gene Mutation Database

When EVE’s uncertain class were retained, how were they treated? Added to the pathogenic or benign sets or split using the EVE gene-specific threshold?

A distance of 11Å seems very large, it is stated that no larger distances were tested due to the danger of capturing protein-specific properties, but how do we know they are not being captured at this threshold?

The second-to-last sentence of “Epistatic and Structure-based features” in the Methods makes no sense to me.

Figure 1 – I like this figure a lot, but I believe that gnomAD should be mentioned by name in the first box as it was the primary source of benign variation.

Figure 2 – Panel A is very information-heavy and there are no navigational aids to help direct the reader.

The coevolution plots on Panel B have an issue with similar colours in the outer Pfam domain circle (DNA-binding domain and Disordered regions look very similar in the P53 plot). The Pfam band is not mentioned at all in the legend, which talks about the outer band being the partners score.

Figure 4 – I was initially confused as to why there were multiple DeMAG bars in each panel. This is explained in the legend of Table 1, but should also be described in Figure 4 so that the figure can be interpreted without reference to the Table 1 legend.

The nature of the error bars in Panel B are not described.

Website

The website has some display issues, in the "Look Up Variants" section, the prediction overlaps the mutation landscape box on my browser. The tables of testing data cannot be scrolled, despite containing more columns than fit on my screen!

Reviewer #3 (Remarks to the Author):

In this manuscript, the authors have discussed a new variant effect predictor (VEPs) based on a supervised classification algorithm (DeMAG). The work's novelty lies in the inclusion of the structural and evolutionary coupling information into the feature list to train the model. The authors have introduced a new feature, "partner's score", which represents a posterior probability belonging to the pathogenic class given the evolutionary score of the residue. The result from the supervised classifier shows that the sensitivity and specificity of the DeMAG model are better than the existing VEPs.

Major issues:

1) This method also has less misclassification error compared to existing methods. Including epistatic information with evolutionary coupling has already been discussed in previous methods like EVMutation, and EVE. Although this model performs better compared to EVE, these results are due to the supervised learning used in DeMAG. Therefore, authors should compare their results to the various supervised methods that have been reported in literature. Just adding a supervised learning to the VEP is not novel. Please compare your work to recent supervised VEPs. doi: 10.1016/j.ajhg.2021.12.007, 10.1016/j.ajhg.2021.08.012, 10.1038/s41467-021-25976-8

2) The authors discussed that structural information is included in the partner's score. However, the residue score is calculated based on the co-evolutionary information. It has been shown that residue scores for close residues and co-evolving residues are highly correlated. Therefore, it is not clear how this feature is considering structural information. Other structural features like pLDDT score and normalized accessible surface area are shown to have negligible feature importance. The structures from AlphaFold2 have large unstructured regions as shown in Figure 2. These regions also include pathogenic mutations which will be missed by Partners score.

Minor issues with the manuscript:

1. Figure 2a captions are too small to read, and can be a separate figure itself.
2. Figure 4a should have error bars
3. In the method section, "Epistatic and structure-based features" subsection, the wrong figure number is referred to in the line "The final alignment coverage..."

Review Response

We thank the reviewers for the thorough evaluation of our manuscript. We believe that we have addressed the concerns about the generalizability of the model by demonstrating its performance on additional 257 established disease genes which have sufficient numbers of variants with high quality diagnostic interpretations. As additional clinical data becomes available, more genes can benefit from the partners score feature, the focal innovation of DeMAG.

With limited independent clinical data, there are challenges in making a fair benchmarking assessment of all VEPs. To overcome this issue, we used variants from the Estonian Biobank, and evaluated the performance of VARITY and ClinPred vs. DeMAG, and updated Table 2 and Fig. 6 with the new results. DeMAG outperforms ClinPred, and VARITY outperforms DeMAG but it covers only 85% of testing variants while DeMAG has 100% coverage. Unfortunately, it is not possible to use the ClinVar test set, because VARITY and ClinPred have used ClinVar variants published until 2021 and 2018, respectively, thus their models have trained on variants that we used in the testing set to benchmark VEPs (type 1 circularity¹). While the ClinPred publication used an older version of ClinVar from 2016², the online ClinPred version uses data from 2018 (<https://sites.google.com/site/clinpred/?pli=1>), which we confirmed by contacting the authors. Instead we have evaluated the performance of ClinPred and VARITY on DeMAG's training set (Response Table 1) and observed DeMAG's superior performance. Since VARITY's authors published the training set, we also compared DeMAG against VARITY training set (on the 257 and 59 genes for which DeMAG has predictions). We observed comparable performance (Response Table 2), considering that DeMAG has never trained on genes other than the 59 ACMG SF v2.0 genes. To sum up, DeMAG performs better on VARITY's training set than VARITY on DeMAG's training set.

Furthermore, we provided a more detailed comparison with EVE, as requested by Reviewer #1 (Response Figure 2, Response Figure 3, Response Figure 4, Supplementary Table 6).

We provided DeMAG software and data as a zip file to the Editor and Reviewers and we will make publicly available the training set at the moment of publication on our web server, as we have done previously with PolyPhen and PolyPhen-2.

Overall, we found that DeMAG outperforms all other tools tested in this set of genes and clinical variants in terms of specificity and coverage (Figure 5).

Below, we have addressed the specific concerns of each reviewer.

Reviewer #1 (Remarks to the Author):

DeMAG is a supervised variant effect predictor for assessing the likely effect of missense variants in 59 disease-related human genes.

The main novel feature of DeMAG that sets it apart from the large number of competing tools in this space is the development of a "partners score", based on the evolutionary interdependence of variants within close proximity within protein structures. This work in particular has been made possible by the recent release of high quality human protein structure predictions by AlphaFold2.

In addition to the described method, the paper is accompanied with an interactive searchable website that provides the resulting predictions on variants in the 59 selected genes.

As with all variant effect predictors in this space, the main challenge is developing a classifier is the identification of sufficiently large and reliable truth sets. In this situation, the main source of "truth" is to be found in curated clinical databases, predominantly ClinVar. Naturally, this greatly limits the amount of high quality training data available, and the authors of this paper have gone to considerable lengths to develop a balanced training data set (and also sufficiently large and independent testing sets).

The performance of DeMAG is compared with several other popular variant effect prediction tools. The author's note that, in testing, many of the assessed tools have comparably good sensitivity, however, except for EVE, most other tools have relatively poor specificity when compared to DeMAG.

The authors also demonstrate the value of the "partners score" in overall classification performance of DeMAG by comparing to a version with the "partners score" removed, with specificity particularly affected.

While the method is novel, interesting and is demonstrated to have good predictive performance compared to a selection of popular competitors, I have a number of reservations about the manuscript as it is currently written. I will detail these issues below. I believe that these issues would need to be addressed if the paper was eventually to be accepted for publication.

The most significant limitation of DeMAG is that it currently only applies to 59 genes, thus has limited apparent generalisability. The other predictors in the comparison typically can be applied to thousands of human genes. Obviously there is a tradeoff to be made here. It is clear that better predictors can be made when they are designed to work on a smaller set of genes. In the extreme limit, a carefully chosen single-gene predictor should have the best performance of all.

Thank you for your comments. We have now measured the generalizability of DeMAG by adding a new analysis which includes additional 257 genes with sufficient clinical diagnostic data to evaluate the partners score feature.

We had initially focused on training a specialized model for the 59 ACMG SF v2.0 genes because of their high clinical relevance and actionability, in an effort to boost performance on this short list. We argue that while general predictors are useful for functional or evolutionary analysis of proteins, they have limited clinical relevance beyond these genes.

Our aim was to develop a tool that can use the wealth of clinical data (e.g., high-quality variants reported in ClinVar, known disease prevalences) to aid clinical-decision making. This allowed training a model with a high-quality balanced training set (60% pathogenic and 40% benign variants, Supplementary Fig. 2d) and a complete feature space (only 6% of missing features annotations). Ensuring balanced classes in the training set as well as avoiding a sparse feature matrix is more challenging when developing a general VEP tool. Indeed, only few genes (4%) have enough high-quality pathogenic and benign labels (**Response Figure 1**), while 14% and 7% of genes have respectively only high-quality benign and pathogenic variants. The fact that variants of the same gene are mutually labeled as either benign or

pathogenic might cause type 2 circularity whenever VEPs use gene-based features or properties of the variants of a specific gene¹.

Response Figure 1

Source: Supplementary Figure 12b and corresponding table (Supplementary Table 5).

The pie chart shows genes distribution based on their associated ClinVar variants. The gray part indicates genes with less than 2 review status stars while the remaining 39% includes genes associated with high-quality variants. Among those, only 4% of genes have at least 5 benign and pathogenic variants.

Given these considerations, we evaluated the generalizability of DeMAG on the genes that have at least 5 benign and 5 pathogenic variants in ClinVar (green). We excluded from the 367 genes the ACMG SF v2.0 genes that we have already studied, and those genes that are more than 1800 residues long (77, 23%), as AlphaFold models are unreliable for very long genes. Overall, 257 genes remain (Supplementary Table 5). We used the same model trained on the 59 ACMG SF genes and evaluated its performance on the 257 new genes. The results demonstrate that DeMAG generalizes well, reaching 91% sensitivity and 85% specificity (Supplementary Table 6).

We provide the above analysis as an example of how to extend our model to other genes via the [DeMAG GitLab page](https://git.mpi-cbg.de/tothpetroczylab/DeMAG/-/blob/main/demag_newgenes/runDeMAG_newgenes.md) (https://git.mpi-cbg.de/tothpetroczylab/DeMAG/-/blob/main/demag_newgenes/runDeMAG_newgenes.md).

Extending a classification algorithm to a wider domain of input cases is usually going to come at a loss of prediction performance. This is especially important when comparing the results of DeMAG to its nearest competitor EVE. For most test metrics, except perhaps coverage, EVE performs very well compared to DeMAG.

However, and most importantly, EVE's publication covers 3219 human genes; a couple of orders of magnitude more than DeMAG. In fact the authors of the DeMAG paper seem reticent to mention this fact, as it is not until deep into the paper that the number of covered genes is mentioned. This is a serious omission. I would expect the number of supported genes to be mentioned in the Abstract and the Introduction of the paper.

Given that the performance of EVE is quite comparable to DeMAG, and that EVE is applicable to a much wider number of human genes, I think the authors must provide a deeper comparison of the two methods, and a stronger argument as to why the "partners score" approach adds value above the method employed in EVE. One is left wondering whether it would be better to add the partners score to EVE, rather than writing a whole new classifier.

We agree that this was unclear in the abstract and introduction, and we have now made it explicit that DeMAG is intended to be a specialized VEP (lines 23-25 and lines 110-111). Our primary motivation is to take advantage of the wealth of diagnostic data available for those genes to develop a tool that outperforms others on this selected list of genes. We have now included the following changes:

“Here, we developed Deciphering Mutations in Actionable Genes (DeMAG), a supervised classifier trained using the extensive diagnostic data available in 59 actionable disease genes (ACMG SF v2.0).”

"[...]we focused on interpreting missense variants in 59 clinically actionable disease genes in the ACMG SF v2.0 list, which we refer to as ACMG SF genes²."

Given that DeMAG is focused on training a model using labeled data in a more limited set of genes, EVE is not necessarily the closest competitor to our method. To our knowledge, EVE's gene-specific threshold actually includes two values: a lower and upper boundary set respectively for the benign and pathogenic class (**Response Figure 1**), which might be useful for stratifying likely pathogenic or likely benign variants. However, variants within those two thresholds are classified as *uncertain*. These thresholds vary across genes and can be set to include different levels of uncertainty.

From the supplementary material of Frazer et al.³:

“In practice, we envision the user of our scores to use this uncertainty metric on a gene- by-gene basis according to the desired precision/recall.”

Response Figure 2

Dots represent ClinVar pathogenic and benign variants (the same as those used for DeMAG benchmarks in Table 1 and Figure 5 presented in the manuscript). The blue dashed line represents the median across the 59 genes of the lower boundary (0.36) for the *uncertain* variants and the red one for the upper boundary (0.64). The black line represents DeMAG's threshold of 0.5. Even if we used genes' specific threshold we would not be able to assign those variants to either classes.

EVE does perform very well compared to DeMAG when we exclude the variants that are classified by EVE as *uncertain*. If we assign the *uncertain* class variants as if EVE was a random predictor then it is clear that DeMAG outperforms EVE (see Table 1 and Figure 5).

VEPs	Sensitivity	Specificity	Accuracy	MCC	AUC	Variants Predicted (n = 1285)	VEP's coverage
DeMAG	0.93	0.82	0.90	0.77	0.96		
SIFT4G	1	0.07	0.68	0.21	0.81	1226	0.954
DeMAG	0.94	0.82	0.89	0.76	0.96		
REVEL	0.99	0.62	0.86	0.69	0.96	1285	1
DeMAG	0.93	0.79	0.89	0.74	0.95		
DEOGEN2	0.91	0.58	0.80	0.53	0.91	964	0.750
DeMAG	0.93	0.79	0.89	0.74	0.95		
PolyPhen2	0.91	0.72	0.85	0.64	0.86	972	0.756
DeMAG	0.94	0.81	0.89	0.76	0.96		
VEST4	0.99	0.64	0.87	0.72	0.95	1280	0.996
DeMAG	0.94	0.81	0.90	0.76	0.96		
M-CAP	1	0.28	0.77	0.44	0.93	1256	0.977
DeMAG ^{a/b}	0.96/0.96	0.81/0.80	0.92/0.91	0.80/0.78	0.97/0.96		
EVE ^{a/b}	0.92 /0.86	0.81/0.73	0.89/0.82	0.73/0.58	0.91/0.89	940/1165	0.731/0.906

^aIf EVE's Uncertain class variants are excluded from the testing set.

^bIf EVE's Uncertain class variants are assigned as if EVE was a random classifier.

Table 1. DeMAG outperforms other VEPs on clinical variants.

Different performance metrics for DeMAG and seven popular variant effect predictors (VEPs). The test set is assembled from the ClinVar database, consisting of both pathogenic (n=852) and benign variants (n=433) submitted after the year 2017 (see Methods). The comparison in pairs guarantees that each predictor is evaluated on all the variants for which a prediction exists. DeMAG has 100% coverage and it is the best performing tool, as it is the most balanced across all the metrics. The comparison with EVE includes two values. The first value indicates the performance on the variants which are predicted by EVE as benign and pathogenic excluding the uncertain class (73% of all variants). The second value includes variants that EVE misclassifies as *uncertain*. We assigned them as if EVE was a random classifier.

While EVE is a general model, and it could be extended to ~3,000 genes, it is limited by the availability of sufficient alignments. To our understanding from Frazer et. al.³ Supplementary Table 5) (https://www.nature.com/https://static-content.springer.com/esm/art%3A10.1038%2Fs41586-021-04043-8/MediaObjects/41586_2021_4043_MOESM5_ESM.xlsx), 1,567 genes (49%) have no scores computed, and the ROC-AUC is only computed for ~1,500 genes. In fact, the AUC values are only high enough for ~1,400 genes, and EVE seems to perform worse than a random classifier for ~50 genes (see **Response Figure 3**).

Response Figure 3.

The distribution of AUC values of EVE predictions. Note that 1567 genes have no values (nan), and only ~1400 genes have AUC values computed.

Additionally, there are important differences that make DeMAG preferable for the set of clinically actionable genes:

1. Simple machine learning vs deep learning. DeMAG is a simple machine learning tool that only uses 13 features, therefore it is more interpretable than a deep learning tool model such as EVE.
2. EVE's low prediction coverage. DeMAG predicts all possible amino acid substitutions for the ACMG SF v2.0 list (59 genes), while EVE does not have predictions for 17% of mutations and 28% are predicted as *uncertain*. Therefore, 45% of substitutions remain without interpretation, while DeMAG has a coverage of 100% in these genes.
3. Poor quality alignments in low complexity and disordered regions. Benign variants tend to occur in disordered or low complexity regions that are difficult to align⁴. Since EVE is based on multiple sequence alignments it might have less or no coverage at all in these regions. This statement is supported by the figure below (**Response Figure 4**) that shows AlphaFold pLDDT metric for all residues of the ACMG SF v2.0 genes. The distribution of missing variants is characterized by a low AlphaFold pLDDT confidence score, which has already been reported as a metric for disorder⁵. Additionally, EVE classifies as uncertain variants some that are actually in structured regions.

Response Figure 4.

4. Uncertain variants misclassified. We included two numbers as performance metrics for EVE on all the columns of Table 1. The first one indicates the performance on 73% variants where EVE provides a prediction. The second number indicates the performance on 91% of variants, that include those predicted as *uncertain* by EVE. Here, since EVE does not provide a prediction for the *uncertain* class, we assigned half of the variants as benign and half as pathogenic, namely as if EVE was a random classifier. In this case, *uncertain* variants are misclassified variants because they are true pathogenic or benign high-quality (at least 2 review status stars) ClinVar variants (**Response Figure 1**). It is evident based on both sets of data that DeMAG shows a better performance than EVE.
5. DeMAG performs better on 257 extended sets of genes. As previously mentioned, we evaluated DeMAG performance on a set of variants from another 257 ClinVar genes and we showed that DeMAG generalizes well without the need of training a new model. We also compared the performance against EVE and demonstrated that DeMAG performs better (see Supplementary Table 6 below).

VEP	Sensitivity	Specificity	Accuracy	MCC	AUC	Variants Predicted	Variants coverage
DeMAG	0.91	0.85	0.88	0.75	0.95	7911	1
DeMAG ^{a/b}	0.93/0.92	0.86/0.83	0.91/0.89	0.79/0.74	0.96/0.95	7911	1
EVE ^{a/b}	0.89/0.81	0.93/0.86	0.90/0.83	0.78/0.63	0.95/0.93	5427/6682	0.69/0.84

^aIf EVE's Uncertain class variants are excluded from the testing set.

^bIf EVE's Uncertain class variants are assigned as if EVE was a random classifier.

Supplementary Table 6. DeMAG generalizes to additional 257 disease associated genes.

The first line shows DeMAG's performance on 257 ClinVar genes with at least 5 benign and 5 pathogenic high-quality (at least 2 review stars) variants. DeMAG has predictions for all 7911 (100%) variants, and reaches 91% sensitivity and 85% specificity. The high performance of DeMAG confirms that it is a generalizable predictor that can be applied to other genes without re-training the model.

Comparison with EVE on the 257 genes: the first numbers (superscript a) refer to the performance on 5427 (69%) variants that EVE predicts as either benign or pathogenic (missing or "uncertain" predictions are excluded). The second numbers (superscript b) refer to the performance on 6682 variants (84%) which include variants that EVE classifies as "uncertain" but are high-quality ClinVar benign or pathogenic variants. The assignment of those variants to either class was random. In both cases, DeMAG outperforms EVE.

It is an interesting suggestion to combine EVE and DeMAG or simply EVE and the partners score, and it certainly deserves attention as a potential future research question. However, we believe it is currently beyond the scope of this paper, especially considering the fundamental differences between these two tools and the complexity in reliably combining them.

While it is pleasing to see that the manuscript is accompanied by an interactive searchable website (<https://demag.org/>) I could not find any way to download the entire DeMAG classifications in one bulk file. Many other classifiers make this information available in bulk. Searching for individual variants is somewhat useful, but it is hard to incorporate this data into other analyses or perform comparisons with other methods without a bulk download. It will not be a large amount of data, so I do not think there is any technical reason why a bulk download is not provided. If such an option is available, I was not able to find it in the interface of the website, and therefore I suggest that such a feature be made prominent.

We are sorry for the inconvenience. We have provided the code and all bulk data as a zipped file for the reviewers which contained all data, scripts and the model, with a readme file. We activated the bulk download option on the website for testing and predictions data. Once the paper is accepted we will enable the download of the training data.

Overall the paper was relatively easy to understand, however, there were many parts where I felt that the writing was vague, unclear or awkwardly phrased. It seemed as though the writing was rushed, and it would benefit from further proof reading. This was particularly apparent in the Supplementary methods section. On this topic I have provided a non-exhaustive list of parts of the paper that would benefit from redrafting.

Thank you for pointing this out. We combed through the manuscript to clarify phrases and improve the readability of the manuscript (e.g., pp.197-202, 252-256, 263-266, 278-280, 310-319, 386-391).

I commend the authors for making their software available as well. This does aid to the reproducibility of the work. However, I would say that I would struggle to reproduce the method from the description in the paper and the Supplementary materials, and I have previous experience implementing variant effect prediction tools.

Thank you, we completely agree and support open software and data sharing. We edited the methods section and the supplementary material. Hopefully, now it will be easier to reproduce our method.

Specific comments about the text of the manuscript are below:

Human gene names should be italicised.

(page 3) This sentence needs clarification: "Indeed, human pathogenic variations appear as neutral substitutions in closely-related orthologous from other species." Surely this is not true for all human pathogenic variations. Is there a citation for this? Also, should it be "orthologs"?

Thank you for catching this misspelling. Yes, it should be orthologs. We added two citations to support the claim and corrected the sentence: **For example, around 10-15% of substitutions in non-human proteins are known to be pathogenic in their human orthologs.**^{26,27}

We added two citations for this Kondrashov et al. (see Figure 3 from Kondrashov et al. below from **Response Figure 5**)⁶ and Sundaram et al. NatGen 2018 (see Figure 1e below from Sundaram et al. from **Response Figure 5**)⁷.

Response Figure 5.

(page 3) "in clinically actionable disease genes in the ACMG SF v2.0 list" you should say how many genes there are in this list. Also this piece of information should be mentioned in the abstract. It is an important detail.

It is indeed an important detail, we added it in the abstract and repeated it in the introduction (see response on page 4).

(page 3) "Under and overdiagnosis as well as patients' psychological burden due to lack of evidence to support variants' pathogenicity may result in increased costs for the healthcare system." This seems quite speculative. It might be true, but is there any data to back up this claim?

Yes, there is data. We have now provided some references for this claim.

- Richter, S. *et al.* Variants of unknown significance in BRCA testing: impact on risk perception, worry, prevention and counseling. *Ann. Oncol.* **24 Suppl 8**, viii69–viii74 (2013).
- Ong, M.-S. & Mandl, K. D. National expenditure for false-positive mammograms and breast cancer overdiagnoses estimated at \$4 billion a year. *Health Aff.* **34**, 576–583 (2015).
- Makhnoon, S., Shirts, B. H. & Bowen, D. J. Patients' perspectives of variants of uncertain significance and strategies for uncertainty management. *J. Genet. Couns.* **28**, 313–325 (2019).

(page 5) "Many existing VEPs do not explicitly attempt to balance sensitivity and specificity." This is a bold far reaching claim that must be supported by evidence. Also the word "many" is vague and imprecise. It reads more like an opinion or conjecture than a fact.

Thank you for pointing this out. We rephrased the sentence: **Several existing VEPs, such as M-CAP and SIFT4G have high sensitivity but low specificity⁸.**

(page 6) "While EVE and DeMAG outperform other tools, their misclassification rate is still high on DMS datasets, which underlies a potential limitation for the clinical application of such data." Is the implication that that DMS data is misleading or the predictions are wrong?

This is an interesting question. DMS datasets are based on functional assays that are only proxies of the phenotypic effect on the organism, in these cases on human diseases. Although, for three genes it was possible to reclassify VUSs based on the results of multiplexed assay experiments⁹, it remains difficult to determine to which extent functional assays are a measure of disease risk or pathogenicity¹⁰. In addition, without enough clinically labeled control variants it is not possible to assess the evidence of functional assays¹¹.

We rephrased the text to avoid unexpected implications:

"The concordance between VEPs and multiplexed assays of variant effects (MAVEs) on clinical data has been frequently reported^{20,52–54}, and both techniques are used to reevaluate VUSs according to ACMG-AMP guidelines⁵⁵. Usually, the agreement between DMS data and VEPs is assessed through correlation coefficients and AUCs⁵⁶, which do not depend on a classification threshold that is needed for clinical decision-making.

We validated DeMAG **as well as other VEPs** against Deep Mutational Scanning (DMS) data as in prior studies^{20,23,53}, using data for 4 genes (*BRCA1*⁵⁴, *TP53*⁵⁷, *PTEN*⁵⁸ and *MSH2*⁵⁹). Most variants in these assays are not yet annotated in ClinVar (51%), and 41% are ClinVar VUSs (Supplementary Fig. 8). Among the 7 VEPs evaluated, DeMAG performed best on *BRCA1* (35% Matthews Correlation Coefficient, MCC) and *PTEN* (27% MCC), while EVE performed better for *TP53* (39% MCC) and *MSH2* (38% MCC) (Supplementary Fig. 8).

The DMS data analysis indicates an overall poor performance of VEPs on functional data, at least on these four genes (Supplementary Fig. 8). Importantly, while sensitivity and ROC-AUC values are high (>80%) for all VEPs tested, the specificity values are worse than the one of a random classifier (Supplementary Fig. 8). This might be due to the high proportion of variants interpreted as functional by DMS data, e.g., for *MSH2*, 92% of single nucleotide missense substitutions are assessed as functional, and classified as pathogenic by VEPs. These results are in agreement with previous studies reporting weak performance of VEPs in predicting beneficial mutations (gain-of-function) in DMS data⁵³, and overall low concordance (Spearman correlation coefficient < 50%) between VEPs and DMS data¹⁸.

(page 6) This sentence is confusing, please clarify: "As the most recent supervised method was published in 2016, none of the VEPs were trained on those newer variants."

We rephrased the sentence as follows in the manuscript: **"We assume that as the most recent supervised method we benchmarked with was published in 2016, none of the VEPs trained on those variants as they were not yet part of the ClinVar database."**

In order to avoid type 1 circularity we built a testing set that contains only variants submitted to the ClinVar database after the year 2017. Since we benchmarked with supervised VEPs published until 2016, they could have not been trained on variants submitted after 2017. We assume that if the variants were submitted to ClinVar after 2017 they were not earlier available in any other database.

(page 8) This comes out of the blue with no mention previously in the paper: "Nevertheless, the comparison between AlphaFold2 and IUPred2A showed how high-quality predictions are still missing for many ordered regions (Supplementary Fig. 6), highlighting how the mystery of protein folding is yet to be understood." This needs some kind of previous description.

We removed the paragraph.

(page 10) "experimental replicas" replicates?
corrected

(page 11) python -> Python
corrected

(page 12) rephrase: "in a connection with only not annotated residues"

Done.

(page 12) "residues-based" -> residue based

We fixed it.

(page 12) this is unclear and imprecise: "thus we combined them and in case of overlap we took the spatially close residues score"

We rephrased the methods.

(page 12) "The best model selected by BIC" I believe this is the first mention of Bayesian Information Criteria (BIC). This probably deserves a bit more explanation, not all readers will be statisticians or data scientists.

We added **Bayesian Information Criteria**.

(page 12) suggest rephrasing "We considered the posterior probability of pathogenicity as the partners score" to something like "We define the partners score to be equal to the posterior probability of pathogenicity"

Thank you for noticing this. We have rephrased the definition.

(page 12) "Eventually" avoid this word in this context. It doesn't suit the scientific description of a method.

We changed the wording throughout the text.

(page 12) rephrase "by dividing the times the t-statistic"

We fixed it.

(page 13) "Since July 2021 we updated our 3D data with AlphaFold2 models" this part of the method is very unclear. If AlphaFold2 models have made substantial changes to the method from previous iterations, then why mention anything before this? Does this not obviate the need to discuss the previous methods?

Since this project started 3 years ago, we have put substantial effort into deriving structures for all residues in the 59 genes, and we performed homology modeling and de novo modeling. While experimentally determined structures covered 28% of residues, our efforts led to ~50% coverage (see part of Suppl. Fig.5. below). When the AlphaFold2 predictions were released in July 2021, our previous work became obsolete. We have now removed the mentioning of the previous work to simplify the methods section and the supplementary material.

Removed from previous Supplementary Figure 5.

(page 20) X axis labels for the comparisons would really help, otherwise reader has to compare colours.

Thank you for this great suggestion. We have added the labels now.

Reviewer #2 (Remarks to the Author):

Overall, this was an interesting study. I particularly liked the idea of the novel ‘partners score’ feature and its predictive ability seemed to be excellent. The paper is well presented, and I have little doubt as to the performance of DeMAG in the 59 clinically actionable genes. My concerns primarily regard the scope of the study, the other VEPs benchmarked against and the availability of the data.

Thank you for appreciating the development of the ‘partners score’ and for your comments.

Major concerns

Study scope: It is mentioned in the discussion that up to 30% of VUS in ClinVar are from ACMG genes. This means that DeMAG cannot make predictions on $\geq 70\%$ of identified VUS. I find it unlikely that many non-specialist variant annotation pipelines (many of which still use SIFT and PolyPhen-2) would add a method which is unable to make predictions in a majority of variants.

We apologize for the wrong number. Actually the ACMG SF v2.0 genes cover 16% of VUSs in ClinVar (the 30% refers to high-quality VUSs which is most probable is not a meaningful distinction in the case of VUSs).

This is an important point that we did not stress enough. The goal of this work is to create a classifier for clinically relevant genes, we have thus changed the abstract to highlight this. The 59 genes that are recommended by the American College of Medical Genetics and Genomics are called “actionable genes”, because preventive diagnosis or treatment is available. Therefore, every time a person undergoes whole exome or genome sequences, variants in these genes are reported as secondary findings, unless the patient opts out¹⁶. Clinical

sequencing reports will notify the patients about potentially pathogenic variants that are determined by well-regulated criteria that include the usage of VEPs¹⁷. We believe that our predictor will be adopted by the medical community for the interpretation of variants in these specific genes.

While general VEPs can make predictions for all genes, their clinical relevance is questionable, as well as their performance since in the absence of ground truth data it is not possible to test (see response to Reviewer #1 and the new Supplementary Figure 12).

The histogram below (from Supplementary Fig. 12a) shows that among genes associated with at least 2-stars reviewed clinical variants (3,626 genes), 50% of the genes have less than 4 variants and 75% less than 11. The data is based on the ClinVar VCF file version 08-12-2022.

Supplementary Figure 12a.

The histogram shows the distribution of high-quality (review status stars ≥ 2) variants across ClinVar genes: 75% of genes have less than 11 variants, 23% have between 11 and 70 variants and the last 2% of genes (not shown in the plot) have between 70 and ~800 variants.

DeMAG is only compared to supervised methods from pre-2016 (plus EVE) to counter the issue of type-1 circularity. Unfortunately this leaves us uncertain about its performance levels against more state-of-the-art predictors (e.g. VARITY or ClinPred). The claim in the abstract that DeMAG out-performs existing VEPs has only been shown to be true for VEPs 8+ years old. As a suggestion, the DMS analysis could be expanded to include some more modern VEPs.

While it is true that we benchmarked "old" VEPs tools, we would like to point out that those are the most popular and widely-used tools (SIFT, PolyPhen, REVEL), with REVEL outperforming the newer meta-predictor ClinPred¹⁵. Furthermore, we compared against EVE which represents one of the state-of-the-art tools for pathogenicity prediction (see benchmarking of EVE against 15 VEPs in Figure 2c of Frazer et. al.³ above by Reviewer #1). As the reviewer correctly points out we can't have a fair comparison with newer tools, thus we decided to compare the test-on-training performance of DeMAG to VARITY and ClinPred

(see **Response Table 1** below). Even if it is not the most accurate comparison, we can appreciate and confirm that DeMAG strength is classifying benign variants.

Since VARITY's authors published the training set, we also compared DeMAG against VARITY training set (on the 257 and 59 genes for which DeMAG has predictions). We observed comparable performance (**Response Table 2**), considering that DeMAG has never trained on genes other than the 59 ACMG SF v2.0 ones. To sum up, DeMAG performs better on VARITY's training set than VARITY on DeMAG's training set.

VEP	Sensitivity	Specificity	Accuracy	MCC	AUC	Predicted variants	Variants coverage
DeMAG	0.90	0.84	0.88	0.74	0.94	11225	1
ClinPred	0.95	0.60	0.82	0.62	0.91	10842	0.97
DeMAG	0.89	0.84	0.87	0.73	0.94	11225	1
VARITY_R	0.86	0.74	0.81	0.60	0.89	9249	0.82

Response Table 1.

Evaluation of ClinPred and VARITY (VARITY_R score was used, results did not change with VARITY_ER) performance on DeMAG training set variants. DeMAG has the highest AUC and specificity compared to both tools. In addition, both tools have lower than 100% coverage of variants.

VEP	Sensitivity	Specificity	Accuracy	MCC	AUC	Predicted Variants	Variants Coverage
DeMAG	0.90	0.83	0.89	0.68	0.94	12419	1
VARITY_rare	0.91	0.87	0.90	0.73	0.96	12397	0.998

Response Table 2.

Evaluation of DeMAG performance on VARITY_R training set variants. Strangely, 22 variants from Q8N726 uniprot id are missing (the webserver also does not support this uniprot id <http://varity.varianteffect.org/>) VARITY_R predictions despite being included in the training set. The training set was downloaded from the github page (<https://github.com/joewuca/varity> - varity_r_input_data.csv).

We cannot really benchmark against VARITY or ClinPred on the ClinVar test dataset, since VARITY was published in September 2021 and therefore it has used the variants of our test set in the training process and likewise ClinPred. The only independent test set is the Estonian Biobank data, where we can truly compare DeMAG and VARITY. See updated Figure 6 and Table 2. VARITY has higher performance on the Estonian data, but DeMAG has higher (100%) variant coverage.

Concerning the analysis on DMS data, we showed that VEPs do not correlate with DMS data (Supplementary Fig. 8). Thus, interesting future research might be focusing on the source of

discrepancies between *in silico* tools and DMS experiments but it is not within the scope of this paper to address these issues (See response to Reviewer #1).

Features for the model other than the partners score were primarily taken from the PolyPhen-2 feature pipeline, many VEPs have used other innovative useful features since PolyPhen-2, it would have been interesting to see if any key features from other VEPs improved performance.

DeMAG uses 13 features. 9 of them have been already used in PolyPhen-2, 3 of them are novel: IUPred score, pLDDT score, and 'partners score' and the last feature, EVmutation, partly answers the reviewer suggestion as this feature is also used in the VARITY model. Furthermore, we manually curated the alignments in order to optimize the coverage of EVmutation (from 37% to 64% of residues) which nevertheless contribute on par with the difference of the PSIC scores calculated for the mutant and the wild-type residue¹⁸.

VARITY has an innovative set of features concerning protein-protein interactions. Since interfaces are under functional constraint, they may tolerate less mutations, and also tend to evolve slower. We hope the conservation scores already account for this effect. Modeling protein-protein interfaces is not a trivial task, it could be done using AlphaFold-Multimer¹⁹. It would be interesting to test this feature in the future.

ClinPred uses population sequencing based allele frequency data, and those features were shown among the most important ones. We did not want to add these features since we have used allele frequency thresholds to define benign variants in our training and test sets.

The amount of hyperparameter optimisation performed seems rather small (3-value grid search across 2 parameters), given the amount of effort put into creating and evaluating the partners score feature, this lack of optimisation for the final model was surprising.

Thank you for the suggestion. We have not seen significant improvement (see Supplementary Table 1) when we fitted the two hyperparameters, shrinkage and interaction depth, presumably because of the small training set. Therefore we did not tune the remaining two hyperparameters.

A lot of care was used to create manually curated multiple sequence alignments to generate the evolutionary couplings that form the basis of the partners score. If EVE or SIFT (or even EVmutation) were run with these high-quality alignments rather than their native ones, perhaps they would perform better?

Yes, indeed, we tried to maximize the coverage of the alignments, which is a non-trivial task outside of well-conserved domains. We could increase residue coverage from 37% to 64% with our “tiling pipeline”. While it is computationally expensive to do it proteome-wide, it would be the preferred method to compute EVmutation or other conservation scores. We agree

with the Reviewer that EVE would also benefit from alignments with more coverage because this will allow the prediction of more variants. EVE can't predict 17% of all possible amino acids substitutions in the ACMG SF v2.0 genes, likely due to lack of sufficiently deep alignments (**Response Figure 4**, above).

There is nowhere to download the full set of predictions! This stumps external benchmarking efforts.

We are sorry for the inconvenience. We have provided the code and all bulk data as a zipped file for the reviewers which contains all data, scripts and the model and readme file. We deactivated the bulk download option on the website until the paper is published. Once the paper is published we will enable download of all training, test and prediction data.

Minor concerns

Citations for “The uncertainty about the pathogenicity of a variant may pose a psychological burden for patients” (10.1093/annonc/mdt312)

Thank you for pointing us to this reference, we have included it in our manuscript.

“Unsupervised methods, such as DeepSequence¹², EVmutation¹³ and EVE¹⁴ are agnostic to variant labels as they infer functional effects from multiple sequence alignment (MSA) alone”
Not all unsupervised methods infer effects from a MSA, for example Eigen.

We changed the sentence to “Several unsupervised methods ...”.

Putatively benign variants is never defined.

Thank you for spotting this missing definition. We refer to putatively benign variants as those common variants from non-human primates polymorphisms as we do not have any functional evidence of the effect of such mutations in humans. In addition, we consider putatively benign variants that come from population data when they support a BS1 classification^{20,21} (MAF greater than the disease prevalence). Those variants are considered benign to train and test the model.

We have clarified the paragraph in the Online Methods subsection Training dataset. To avoid any confusion we now only refer to benign variants throughout the manuscript text.

Human Mutation Gene Database -> Human Gene Mutation Database

corrected

When EVE's uncertain class were retained, how were they treated? Added to the pathogenic or benign sets or split using the EVE gene-specific threshold?

Previously, we have treated the *uncertain* class as misclassified. We have realized that a more accurate treatment is to assign them as if EVE was a random classifier for these variants (50% benign and 50% pathogenic). We added the updated performance metrics in Table 1 and Fig. 5. To our knowledge there is no single gene-specific threshold that we could use to further classify EVE's *uncertain* variants as either pathogenic or benign. To our understanding, each gene has an upper (~ 0.6) and lower boundary (~ 0.3), above and below which variants are classified as pathogenic or benign. The *uncertain* class variants have a score between ~ 0.3 and ~ 0.6 , therefore even if we used those specific thresholds the variants with scores in between 0.4 and 0.6 will still lack a clear clinical interpretation (**Response Figure 2**).

A distance of 11\AA seems very large, it is stated that no larger distances were tested due to the danger of capturing protein-specific properties, but how do we know they are not being captured at this threshold?

Thank you for raising this concern. In general, protein-specific features bias classifiers to predict properties of a gene than variant-specific ones¹. In the context of clinically actionable genes and in particular of the ACMG SF v2.0 we showed that the distribution of pathogenic and benign variants is highly unbalanced (Supplementary Fig. 2d): 80% pathogenic and 20% benign. In this case, using gene-specific properties will only worsen the circularity problem (type 2 circularity¹) for which more variants will be labeled as pathogenic because they occur in genes that harbor mostly pathogenic variants (e.g., because they are more studied than others). DeMAG reduces this bias by curating a training set that consists of 60% pathogenic and 40% benign variants (Supplementary Fig. 2d) and it does not include any gene-specific feature, thus it is less subject to gene-specific biases.

In addition, we do not observe that the performance of the 3D score improves as the \AA distance increases (Supplementary Fig. 6b), which we would expect if we captured protein-specific properties that introduce circularity. Moreover, we calculated the radius of gyration for the AlphaFold 3D models which can be considered as a metric for protein compactness²². The plot below shows that most of the proteins have a radius of gyration greater than 30, thus we can consider 11\AA as a threshold that does not capture protein-specific characteristics (**Response Figure 8**).

Response Figure 8.

The plot shows the radius of gyration for the ACMG SF v2.0 genes AlphaFold2 3D models.

The second-to-last sentence of “Epistatic and Structure-based features” in the Methods makes no sense to me. We have now rephrased the entire section of the Methods.

Figure 1 – I like this figure a lot, but I believe that gnomAD should be mentioned by name in the first box as it was the primary source of benign variation.

Thank you for the comment. We added gnomAD to Figure 1.

Figure 2 – Panel A is very information-heavy and there are no navigational aids to help direct the reader. The coevolution plots on Panel B have an issue with similar colours in the outer Pfam domain circle (DNA-binding domain and Disordered regions look very similar in the P53 plot). The Pfam band is not mentioned at all in the legend, which talks about the outer band being the partners score.

We have split Figure 2 into two figures (Figure 2 and Figure 3) and improved the clarity of the colors and the legend.

Figure 4 – I was initially confused as to why there were multiple DeMAG bars in each panel. This is explained in the legend of Table 1, but should also be described in Figure 4 so that the figure can be interpreted without reference to the Table 1 legend.

The nature of the error bars in Panel B are not described.

Thank you for pointing this out. We fixed the Figure legend and added the missing information about the error bars.

Website

The website has some display issues, in the “Look Up Variants” section, the prediction overlaps the mutation landscape box on my browser. The tables of testing data cannot be scrolled, despite containing more columns than fit on my screen!

Thank you for noticing these issues. We have fixed the display of figures and tables and tried to make the website more dynamic and browser-independent. We suggest using Safari or Google Chrome.

Reviewer #3 (Remarks to the Author):

In this manuscript, the authors have discussed a new variant effect predictor (VEPs) based on a supervised classification algorithm (DeMAG). The work's novelty lies in the inclusion of the structural and evolutionary coupling information into the feature list to train the model. The authors have introduced a new feature, "partner's score", which represents a posterior probability belonging to the pathogenic class given the evolutionary score of the residue. The result from the supervised classifier shows that the sensitivity and specificity of the DeMAG model are better than the existing VEPs.

Major issues:

1) This method also has less misclassification error compared to existing methods. Including epistatic information with evolutionary coupling has already been discussed in previous methods like EVMutation, and EVE. Although this model performs better compared to EVE, these results are due to the supervised learning used in DeMAG. Therefore, authors should compare their results to the various supervised methods that have been reported in literature. Just adding a supervised learning to the VEP is not novel. Please compare your work to recent supervised VEPs. doi: 10.1016/j.ajhg.2021.12.007, 10.1016/j.ajhg.2021.08.012, 10.1038/s41467-021-25976-8

Thank you for your comments. While it has been shown that epistatic information improves assessment of variants effects^{3,23,24}, the partner score is a new type of feature, since it includes information of known variants. We were inspired by the previous methods and tested the hypothesis if epistatic residues tend to share the same phenotypic effect (Supplementary Figure 6a and b). Indeed, residues that are co-evolving or close in 3D space tend to share variants with the same phenotypic effect (i.e. pathogenic or benign label). Therefore, we built the partners score feature (see Methods).

We would like to note that StrVCTVRV (doi: 10.1016/j.ajhg.2021.12.007) is a pathogenicity predictor of Structural variants (SVs; genomic variants larger than 50 base pairs), and it is not applicable to Single Nucleotide Variants (SNVs).

ECnet is a promising method that was trained and benchmarked on DMS datasets²⁴. Comparison on 39 DMS dataset (of 33 genes) showed that it outperforms all supervised and unsupervised methods. The caveat is that performance was assessed as rank correlation (Spearman correlation coefficient) or as the ability to identify variants that are better than the wild type (ROC-AUC). To our understanding, the authors do not provide a score threshold that would identify deleterious (and potential pathogenic) mutations. They focus on protein engineering applications and not on clinical variant interpretation. Additionally, they do not provide precomputed scores for the genes of interest in our paper. Therefore, we cannot directly compare DeMAG to ECNet.

We extended the benchmarking and compared DeMAG to two recently published meta-predictors, VARITY (published in 2021)²⁵ and ClinPred (published in 2018)². Since both

of these tools have used ClinVar variants of our test set for their training, we could not use the same comparison as with other methods. We could only compare performance by “testing on training data” (see Response to Reviewer #2), and on the variants from Estonian Biobank (Table 2 and Fig. 6). As it was previously observed¹⁵ on the Deciphering Developmental Disorders (DDD) dataset, REVEL outperforms ClinPred. Indeed, ClinPred has a very high misclassification rate on the Estonian Biobank data (73%, Table 2 and Fig. 6). On the other hand, VARIETY has better performance than DeMAG (30% vs 35%) but it provides predictions for 85% of the variants, while DeMAG reaches 100% coverage (Table 2).

2) The authors discussed that structural information is included in the partner's score. However, the residue score is calculated based on the co-evolutionary information. It has been shown that residue scores for close residues and co-evolving residues are highly correlated. Therefore, it is not clear how this feature is considering structural information.

We were indeed not clear explaining how the partners score combined co-evolutionary and structural information. As the Reviewers states, spatially close and co-evolving residue pairs tend to overlap²⁶. Thus, there is a large overlap between “partners” defined by spatial proximity or co-evolution (Supplementary Fig. 6c). Therefore, we decided to take the union of all partners. In other words, partners are either spatially close or co-evolving residue pairs.

We added the following sentence on page 13:

The residue pairs that co-evolve and are contacting in 3D space overlap and as a consequence the partners scores based on pairs defined by co-evolution and 3D structure correlate (Supplementary Fig. 7c). Therefore, we combined them and took the union of all scores to increase coverage of position. In case of overlap, when both scores were available for a position, we chose the score based on the spatially close residue pairs.

Other structural features like pLDDT score and normalized accessible surface area are shown to have negligible feature importance. The structures from Alphafold2 have large unstructured regions as shown in Figure 2. These regions also include pathogenic mutations which will be missed by Partners score.

It is true, that the feature importance of pLDDT score, ASA and IUPred disorder score is much lower than the partners score for example, but removing those features leads to lower specificity (84% vs 85%). There are of course pathogenic mutations in unstructured regions (10% of pathogenic mutations in DeMAG training set occur in disordered regions i.e. IUPred score ≥ 0.5). The partners score is agnostic to the structural state or the pLDDT score, and it should capture co-evolution based partners in unstructured regions.

Minor issues with the manuscript:

1. Figure 2a captions are too small to read, and can be a separate figure itself.

Thank you for this suggestion. We splitted Figure 2 into two separate figures now.

2. Figure 4a should have error bars

Yes, you are correct. We have now added the error bars.

3. In the method section, "Epistatic and structure-based features" subsection, the wrong figure number is referred to in the line "The final alignment coverage..."

Thank you for pointing this out, we fixed this mistake.

References

1. Grimm, D. G. *et al.* The evaluation of tools used to predict the impact of missense variants is hindered by two types of circularity. *Hum. Mutat.* **36**, 513–523 (2015).
2. Alirezaie, N., Kernohan, K. D., Hartley, T., Majewski, J. & Hocking, T. D. ClinPred: Prediction Tool to Identify Disease-Relevant Nonsynonymous Single-Nucleotide Variants. *Am. J. Hum. Genet.* **103**, 474–483 (2018).
3. Frazer, J. *et al.* Disease variant prediction with deep generative models of evolutionary data. *Nature* **599**, 91–95 (2021).
4. Toth-Petroczy, A. *et al.* Structured states of disordered proteins from genomic sequences HHS Public Access ETOC blurb. *Cell* **167**, 158–170 (2016).
5. Jumper, J. *et al.* Highly accurate protein structure prediction with AlphaFold. *Nature* (2021) doi:10.1038/s41586-021-03819-2.
6. Kondrashov, A. S., Sunyaev, S. & Kondrashov, F. A. Dobzhansky–Muller incompatibilities in protein evolution. *Proc. Natl. Acad. Sci. U. S. A.* **99**, 14878 (2002).
7. Sundaram, L. *et al.* Predicting the clinical impact of human mutation with deep neural networks. *Nat. Genet.* **50**, 1161–1170 (2018).
8. Cubuk, C. *et al.* Clinical likelihood ratios and balanced accuracy for 44 in silico tools against multiple large-scale functional assays of cancer susceptibility genes. *Genet. Med.* **23**, 2096–2104 (2021).
9. Fayer, S. *et al.* Closing the gap: Systematic integration of multiplexed functional data resolves variants of uncertain significance in BRCA1, TP53, and PTEN. *Am. J. Hum. Genet.* **108**, 2248–2258 (2021).
10. Fowler, D. M. & Fields, S. Deep mutational scanning: a new style of protein science. *Nat. Methods* **11**, 801–807 (2014).
11. Brnich, S. E. *et al.* Recommendations for application of the functional evidence PS3/BS3 criterion using the ACMG/AMP sequence variant interpretation framework. *Genome Med.* **12**, 3 (2019).
12. Reeb, J., Wirth, T. & Rost, B. Variant effect predictions capture some aspects of deep mutational scanning experiments. *BMC Bioinformatics* **21**, 107 (2020).
13. Livesey, B. J. & Marsh, J. A. Using deep mutational scanning to benchmark variant effect

- predictors and identify disease mutations. *Mol. Syst. Biol.* **16**, 1–12 (2020).
14. Livesey, B. J. & Marsh, J. A. Interpreting protein variant effects with computational predictors and deep mutational scanning. *Dis. Model. Mech.* **15**, (2022).
 15. Gunning, A. C. *et al.* Assessing performance of pathogenicity predictors using clinically relevant variant datasets. *J. Med. Genet.* **58**, 547–555 (2021).
 16. Loud, J. T. *et al.* Research participant interest in primary, secondary, and incidental genomic findings. *Genet. Med.* **18**, 1218–1225 (2016).
 17. Green, R. C. *et al.* ACMG recommendations for reporting of incidental findings in clinical exome and genome sequencing. *Genet. Med.* **15**, 565–574 (2013).
 18. Sunyaev, S. R. *et al.* PSIC: Profile extraction from sequence alignments with position-specific counts of independent observations. *Protein Eng.* **12**, 387–394 (1999).
 19. Evans, R. *et al.* Protein complex prediction with AlphaFold-Multimer. *bioRxiv* 2021.10.04.463034 (2022) doi:10.1101/2021.10.04.463034.
 20. Amendola, L. M. *et al.* Performance of ACMG-AMP Variant-Interpretation Guidelines among Nine Laboratories in the Clinical Sequencing Exploratory Research Consortium. *Am. J. Hum. Genet.* **98**, 1067–1076 (2016).
 21. Richards, S. *et al.* Standards and guidelines for the interpretation of sequence variants: a joint consensus recommendation of the American College of Medical Genetics and Genomics and the Association for Molecular Pathology. *Genet Med* (2015) doi:10.1038/gim.2015.30.
 22. Lobanov, M. Y., Bogatyreva, N. S. & Galzitskaya, O. V. Radius of gyration as an indicator of protein structure compactness. *Mol. Biol.* **42**, 623–628 (2008).
 23. Hopf, T. A. *et al.* Mutation effects predicted from sequence co-variation. *Nat. Biotechnol.* **35**, 128–135 (2017).
 24. Luo, Y. *et al.* ECNet is an evolutionary context-integrated deep learning framework for protein engineering. *Nat. Commun.* **12**, 1–14 (2021).
 25. Wu, Y., Li, R., Sun, S., Weile, J. & Roth, F. P. Improved pathogenicity prediction for rare human missense variants. *Am. J. Hum. Genet.* **108**, 1891–1906 (2021).
 26. Marks, D. S. *et al.* Protein 3D structure computed from evolutionary sequence variation. *PLoS One* **6**, e28766 (2011).

REVIEWERS' COMMENTS

Reviewer #1 (Remarks to the Author):

This paper has already been through one round of reviews, so I won't recapitulate it again.

Overall the paper has benefitted from the revision and incorporation of feedback from reviewers.

I think DeMAG is an interesting and potentially useful tool for variant classification, and I enjoyed reading the paper and looking into the tool.

While the paper is much improved, I still find that there are many places where the quality of the presentation could be improved. The paper would likely benefit from more proof reading and editing to refine the exposition. Despite that, I think this will be a valuable contribution to the literature and research community.

Specific comments about the submitted version are below:

- It would be useful for the authors to compare their approach to that employed in "mCSM: predicting the effects of mutations in proteins using graph-based signatures. Pires DE, Ascher DB, Blundell TL (2014) Bioinformatics 30 (3):335-342."

- Abstract: "ACMG SF v2.0", should be expanded to say what it is.

- Abstract: not sure why "(AlphaFold)" is mentioned at this point.

- Abstract: "predictions for 316 clinically" -> "predictions for variants in 316 clinically" (ideally say what type of variants)

- The abstract should say what kind of variants can be classified by DeMAG. This is an important detail, and it isn't until about page 4 that SNVs are mentioned. What about indels and other kinds of variants? Presumably those are not handled by DeMAG. Why is this not mentioned at the start?

- Page 2: Unnecessary comma: "These are termed, compensated pathogenic deviations"

- Page 3: "AlphaFold2 predictions" -> "AlphaFold2 3D protein structure predictions, because many disease-related genes do not have an empirically resolved crystal structure"

- page 3: "We developed DeMAG (Deciphering Mutations in Actionable Genes)" repeat of previous explanation of the name on the same page.

-page 3: "Overall, DeMAG used only 13 features," what does the use of only imply here? Was some kind of feature reduction method applied, thus reducing down to 13, or is it merely noted to indicate that only a relatively few features are needed?

- page 3 "Then, we trained a machine learning model". Is this a new paragraph? It seems unusual to start a new paragraph with "Then".

- page 4 "We supplemented that", maybe "We supplemented these"? since you are talking about a collection of variants?

- page 4 "in Human Gene Mutation Database" -> "in the Human Gene Mutation Database"

- page 4 "The last source of pathogenic mutations included all disease-causing 157 mutations from UniProtKB. In total, the pathogenic class consisted of 6,713 unique pathogenic 158 mutations (Fig.1 and Supplementary Fig. 2a)." Suddenly this paragraph starts talking about pathogenic variants. However, the start of the paragraph only talks about a "high-quality training set". Presumably such a training set would have both pathogenic and benign variants in it? So why does the paragraph end by only talking about pathogenic variants? If we read the next paragraph we can see that it is about benign variants, however we should not have to read ahead to find out what the previous paragraph was about.

- page 4 "pathogenic mutations" elsewhere the terminology "variants" are used instead of mutations. I think "variants" is probably better throughout the paper for consistency.

- page 4 "greater than the associated disease prevalence". Sounds reasonable, but how reliable are the statistics on disease prevalence for all the diseases considered by DeMAG? Also, what happens when a

variant is implicated in multiple diseases/phenotypes? Which prevalence figure do you use? Perhaps, to be conservative, you choose the most frequent disease prevalence from the list of alternatives? Maybe this situation never occurs? Okay, I see that some of this is addressed in the Supplementary Methods. Perhaps the details in the main manuscript could be updated to reflect what happens in the absence of prevalence information or other ambiguous scenarios?

- page 4 "Overall, we have a relatively balanced training set," This doesn't sound particularly scientific. Did you investigate what would happen to DeMAG if you used a truly balanced training set (i.e. you picked exactly the same number of pathogenic and benign variants)? Would that make a difference?

- page 5 "co-evolving, share the same phenotypic effect". I can't parse this part of the sentence. I think there is a missing connecting word somewhere.

- page 5 "residues positions" -> "residue positions"

- page 5 "residues positions of the low-complexity region and disordered region are characterized by low partners scores, while residue positions of the DNA-binding domain has overall high scores (Fig. 3a)." When making a claim of association, it may be worth performing a statistic test and stating the result in the paper. In this instance I can't easily see from Figure 3a the association being claimed in the paper. Likewise for "In addition, we observed that the MSH6 ATP binding site has a high partners score (Fig. 3b)." Surely this statement can be made more rigorous?

- page 5 "Our recommended DeMAG score threshold to interpret a variant as pathogenic is 0.5." This is good to know, but this sentence just comes out of the blue, and doesn't really flow with the rest of the paragraph.

- page 5 "In addition, a structural feature, i.e., the normalized accessible surface area, is contributing at least as much as other conservation-based features, e.g., PSIC score." This sentence needs rephrasing. I suggest avoiding the use of "i.e." and "e.g." in this sentence. They are unnecessary and only serve to make it more complicated than it needs to be.

- page 5 "with different proportions of pathogenic mutations" What does "different proportions" mean here?

- page 5 "the improvement at the gene level is more challenging to assess." Can you say why?

- page 6 "DeMAG stood out from all the other predictors, reaching the highest specificity, accuracy, and MCC values" Maybe state the values in the text so readers don't have to go looking at the table to find out?

- page 10. Ideally state the URLs where the datasets were downloaded from.

- page 10. "VUSs label" -> "VUS label"

- page 10. "release 2021_01" was an underscore used on purpose? Previously dates used a dot as the separator.

- page 10. "but only Chimpanzee and Bonobo species were considered" ... because?

- page 11. "the correlation pattern among the replica" -> "... replicas"

- page 11. "we downloaded the predictions from the web server" on what date?

- page 13 "gaussian" -> "Gaussian"

- page 13 "We have downloaded the structure files from ..." on what date?

- page 14 "Missing information is thus treated as information" suggest rephrasing this.

- page 15. "pLDDT" might need to expand what that means.

- page 17. Figure 3. I don't find these sub-figures easy to interpret. Is there a better, simpler way to show the important information being conveyed here? The colours on the 3D protein models are particularly hard to see. I also find the Circos plots hard to understand, especially for MSH6.

Reviewer #2 (Remarks to the Author):

I think the reviewers have done a good job of and put a lot of effort into addressing my specific comments. The method looks interesting, the results seem promising, so I think it is worth publishing. I am somewhat skeptical that DeMAG really is superior to existing predictors, as it is always difficult to fully trust authors' evaluations of their own method. However, the best thing to do is publish the method and see how it performs in independent benchmarking.

Reviewer #3 (Remarks to the Author):

Authors have addressed all the comments raised in the initial review.

Response to Reviewer's comments

Reviewer #1 (Remarks to the Author):

This paper has already been through one round of reviews, so I won't recapitulate it again.

Overall the paper has benefitted from the revision and incorporation of feedback from reviewers.

I think DeMAG is an interesting and potentially useful tool for variant classification, and I enjoyed reading the paper and looking into the tool.

While the paper is much improved, I still find that there are many places where the quality of the presentation could be improved. The paper would likely benefit from more proof reading and editing to refine the exposition. Despite that, I think this will be a valuable contribution to the literature and research community.

We thank the reviewer for his/her enthusiasm for the manuscript, and appreciate his/her attention with respect to the manuscript's text and presentation. We addressed the comments and improved the quality of the presentation.

Specific comments about the submitted version are below:

- It would be useful for the authors to compare their approach to that employed in "mCSM: predicting the effects of mutations in proteins using graph-based signatures. Pires DE, Ascher DB, Blundell TL (2014) *Bioinformatics* 30 (3):335-342."

We thank the reviewer for the suggestion. It would have been an interesting comparison. Unfortunately, the webserver link is not accessible anymore, and we cannot use this method.

- Abstract: "ACMG SF v2.0", should be expanded to say what it is.

Done.

- Abstract: not sure why "(AlphaFold)" is mentioned at this point.

We removed it from the abstract.

- Abstract: "predictions for 316 clinically" -> "predictions for variants in 316 clinically" (ideally say what type of variants)

We edited with the following:

We provide our tool and predictions for **all missense variants in 316 clinically actionable disease genes**.

- The abstract should say what kind of variants can be classified by DeMAG. This is an important detail, and it isn't until about page 4 that SNVs are mentioned. What about indels and other kinds of variants? Presumably those are not handled by DeMAG. Why is this not mentioned at the start?

Thank you for pointing it out. We mentioned in the third last paragraph (line 112) that we focus on the interpretation of missense variants. We added this in the abstract:

Here, we develop Deciphering Mutations in Actionable Genes (DeMAG), a supervised classifier for **missense variants** trained using extensive diagnostic[...].

- Page 2: Unnecessary comma: "These are termed, compensated pathogenic deviations"

Removed.

- Page 3: "AlphaFold2 predictions" -> "AlphaFold2 3D protein structure predictions, because many disease-related genes do not have an empirically resolved crystal structure"

Added.

- page 3: "We developed DeMAG (Deciphering Mutations in Actionable Genes)" repeat of previous explanation of the name on the same page.

We decided to repeat it at the beginning of the results section in case a reader starts reading the manuscript skipping the introduction.

-page 3: "Overall, DeMAG used only 13 features," what does the use of *_only_* imply here? Was some kind of feature reduction method applied, thus reducing down to 13, or is it merely noted to indicate that only a relatively few features are needed?

We wished to highlight that relatively few features are needed. In the era of deep learning methods and of machine learning models with abundant features (e.g., VARITY trains with 43 features, M-CAP with more than 298 features), we find important to highlight that DeMAG reaches high performance with *only* 13 features.

- page 3 "Then, we trained a machine learning model". Is this a new paragraph? It seems unusual to start a new paragraph with "Then".

Removed.

- page 4 "We supplemented that", maybe "We supplemented these"? since you are talking about a collection of variants?

We changed with "We supplemented **pathogenic variants**".

- page 4 "in Human Gene Mutation Database" -> "in *_the_* Human Gene Mutation Database"

Added.

- page 4 "The last source of pathogenic mutations included all disease-causing 157 mutations from UniProtKB. In total, the pathogenic class consisted of 6,713 unique pathogenic 158 mutations (Fig.1 and Supplementary Fig. 2a)." Suddenly this paragraph starts talking about pathogenic variants. However, the start of the paragraph only talks about a "high-quality training set". Presumably such a training set would have both pathogenic and benign variants in it? So why does the paragraph end by only talking about pathogenic variants? If we read the next paragraph we can see that it is about benign variants, however we should not have to read ahead to find out what the previous paragraph was about.

Thank you. This is how we rephrased the paragraph:

In order to curate a high-quality training set, we considered several independent sources of SNVs and set strict criteria to retain only high-quality variants. We included high-quality ClinVar **benign and pathogenic** variants with a review status of at least 'two stars', namely variants labeled with no conflicts between all submitters or reviewed by expert panels or practice guidelines. We supplemented **pathogenic variants** with variants which have previously been described in the medical literature in **the** Human Gene Mutation Database (HGMD)⁴⁰, that have not yet been observed in ClinVar (Supplementary Fig. 4). The last source included all disease-causing mutations from UniProtKB. In total, the pathogenic class consisted of 6,713 unique pathogenic mutations (Fig.1 and Supplementary Fig. 2a).

In addition to ClinVar, we collected benign variants[...].

- page 4 "pathogenic mutations" elsewhere the terminology "variants" are used instead of mutations. I think "variants" is probably better throughout the paper for consistency.

Indeed, the terms, mutation and variant are used interchangeably, the clinical field preferring variant over mutations. We decided to keep the term "disease-causing mutations in UniProtKB" on page 4, since UniProtKB uses this term. And we changed the last sentence to variants for consistency.

- page 4 "greater than the associated disease prevalence". Sounds reasonable, but how reliable are the statistics on disease prevalence for all the diseases considered by DeMAG? Also, what happens when a variant is implicated in multiple diseases/phenotypes? Which prevalence figure do you use? Perhaps, to be conservative, you choose the most frequent disease prevalence from the list of alternatives? Maybe this situation never occurs? Okay, I see that some of this is addressed in the Supplementary Methods. Perhaps the details in the main manuscript could be updated to reflect what happens in the absence of prevalence information or other ambiguous scenarios?

Thank you for noticing this. Prevalence values were considered from Orphanet and MedlinePlus. Supplementary Data 1 contains for each phenotype the source from which the disease prevalence was collected. Indeed, we did exactly as you assumed, e.g., Familial hypercholesterolemia is associated with a disease prevalence of 1-9 / 1 000 000 (according to Orphanet) and 1/200 (according to MedlinePlus), we chose 1/200.

We added those details in the Methods section:

Disease prevalence values were collected from Orphanet and MedlinePlus (<https://www.orpha.net>, <https://medlineplus.gov/>). In the case of multiple disease prevalence values associated to a phenotype we chose the highest value, according to a conservative strategy, and when unavailable, a MAF filter greater than 0.5% was applied (Supplementary Data 1).

- page 4 "Overall, we have a relatively balanced training set," This doesn't sound particularly scientific. Did you investigate what would happen to DeMAG if you used a truly balanced training set (i.e. you picked exactly the same number of pathogenic and benign variants)? Would that make a difference?

We did not investigate this exact scenario since the ratio of pathogenic and benign variants varies greatly between genes (FigS2d). It was already a challenge to design the crossvalidation folds with the constraints of leaving out genes while keeping the balance close to 50%.

- page 5 "co-evolving, share the same phenotypic effect". I can't parse this part of the sentence. I think there is a missing connecting word somewhere.

We removed the comma:

We designed a novel feature called the partners score based on the observation that partner residues that are connected, either because they are close in 3D proximity or because they are co-evolving, share the same phenotypic effect (Supplementary Fig. 6a).

- page 5 "residues positions" -> "residue positions"

Done.

- page 5 "residues positions of the low-complexity region and disordered region are characterized by low partners scores, while residue positions of the DNA-binding domain has overall high scores (Fig. 3a)." When making a claim of association, it may be worth performing a statistic test and stating the result in the paper. In this instance

I can't easily see from Figure 3a the association being claimed in the paper. Likewise for "In addition, we observed that the MSH6 ATP binding site has a high partners score (Fig. 3b)." Surely this statement can be made more rigorous?

Thank you for the comment and the careful reading of our manuscript. We agree that in order to make any claim of association a statistical test should be performed. We added the citation to Supplementary Fig. 13 which shows that the Partners score is significantly higher in Pfam domain regions. To satisfy both the Reviewer and our curiosity we did the same test considering the disorder score and we obtained the same result.

The second statement regarding the MSH6 ATP binding site is trickier to assess. In general, sites' annotation is quite sparse, 70% of annotations are missing in DeMAG training set. Overall, the partners score within sites is significantly lower than outside sites (Wilcoxon test, alternative hypothesis "smaller" and p value <0.01). Nevertheless, those specific residues are associated to ClinVar VUS variants while they are considered LOF by experimental works and they are predicted pathogenic by DeMAG. If we could consider the assay has a "well established" assay this would support a PS3 evidence ("Well-established *in vitro* or *in vivo* functional studies supportive of a damaging effect on the gene or gene product") and PM1 ("Located in a mutational hot spot and/or critical and well-established functional domain without benign variation") evidence according to the ACMG guidelines which will support a likely pathogenic interpretation. We changed the text saying:

In addition, we observed that the MSH6 ATP binding site has a partners score **greater than 0.6** (Fig. 3b). The role of the ATP binding site of the MSH2-MSH6 heterodimer is crucial for DNA mismatch repair (MMR) competency: mutations of the lysine residue in the MSH6 Walker A motif are complete loss of function mutations *in vivo* in *S. cerevisiae*⁴⁹. Moreover, all 14 mutations (G1134[A,R,E,V], P1135A, N1136D, M1137[T,V], G1138R, G1139[D,C,V], S1141[C,P]) in this site are ClinVar VUSs, with no definitive clinical interpretation, **while they are predicted pathogenic by DeMAG**.

- page 5 "Our recommended DeMAG score threshold to interpret a variant as pathogenic is 0.5." This is good to know, but this sentence just comes out of the blue, and doesn't really flow with the rest of the paragraph.

This is how we rephrased the sentence:

Overall and at the single gene level, DeMAG has a balanced sensitivity and specificity (Fig. 4a and c), which contributes to setting the threshold to 0.5 to interpret a variant as pathogenic.

- page 5 "In addition, a structural feature, i.e., the normalized accessible surface area, is contributing at least as much as other conservation-based features, e.g., PSIC score." This sentence needs rephrasing. I suggest avoiding the use of "i.e." and "e.g." in this sentence. They are unnecessary and only serve to make it more complicated than it needs to be.

Thank you, indeed those abbreviations are redundant and worsen the readability of the sentence. We removed them.

- page 5 "with different proportions of pathogenic mutations" What does "different proportions" mean here?

Thank you for the comment. "different proportions" refer to the wide range of pathogenic variants' ratio within genes. We added the reference to another supplementary figure to improve understanding of the statement.

- page 5 "the improvement at the gene level is more challenging to assess." Can you say why?

Thank you for your question. The statement the reviewer cites refers to the paragraph where we show the results concerning the added value of including epistatic and structural features to the DeMAG model. In general, it is possible to assert that the high and balanced performance of DeMAG is respected at the single gene level (Fig. 4a). The contribution of those new features seems to partially depend on the pathogenic-benign ratio of training variants within genes, as we showed in Supplementary Figs. 7 and 11 and discussed in the lines 369-382. To better exemplify this result, let's consider the partners feature as representative of the epistatic and structural

features, since it has the greatest importance among all features (Fig. 4b). For genes with extreme unbalanced pathogenic-benign ratio, e.g., FBN1 93% pathogenic mutations, APOB 87% of benign mutations, the partners score will impute most connected residue positions with the dominant label, namely pathogenic for FBN1 and benign for APOB. This results in an increase in the dominant metric, namely sensitivity for FBN1 and specificity for APOB, and a decrease in the other one.

Nevertheless, there are genes with more than 50% of pathogenic mutations whose sensitivity does not increase with the partners score (Fig. 11a) and we have not identified explanatory features. Those genes have different training set variants ranging from 14 to 358 variants as well as different gene lengths from 166 to 1935 residues. On the other hand, there is only one gene with more than 50% benign mutations, TMEM43, whose specificity decreases with the partners score. There are 24 training set variants for this gene and the particular misclassified 5 benign residue positions seem conserved and in structured regions despite having a low partners score. These might be the reasons why we observed such misclassifications.

To conclude, there are not clear patterns of misclassification, thus a scrupulous analysis gene by gene could help in the interpretation. We believe that other VEP tools should report gene-specific metric as well to support the common effort to aid variants interpretation.

- page 6 "DeMAG stood out from all the other predictors, reaching the highest specificity, accuracy, and MCC values" Maybe state the values in the text so readers don't have to go looking at the table to find out?

It is not possible to report the single values in the text because there aren't single performance values. We benchmarked DeMAG in pairs, therefore, specificity (sensitivity, accuracy, etc.) values are as many as the number predictors we compared against which is 7.

- page 10. Ideally state the URLs where the datasets were downloaded from.

We added the URLs of the training data both in the Methods section and in the Data Collection of the Reporting Summary. In addition we have provided the final training set we used to train DeMAG, with the exception of HGMD variants because they are license-protected.

- page 10. "VUSs label" -> "VUS label"

Removed.

- page 10. "release 2021_01" was an underscore used on purpose? Previously dates used a dot as the separator.

Thank you. The underscore is the preferred format according to UniProt, but we changed with the dot for consistency with previous dates format.

- page 10. "but only Chimpanzee and Bonobo species were considered" ... because?

We added:

but only Chimpanzee and Bonobo species were considered **as the most closely related apes to humans.**

- page 11. "the correlation pattern among the replica" -> "... replicas"

Corrected.

- page 11. "we downloaded the predictions from the web server" on what date?

Added.

- page 13 "gaussian" -> "Gaussian"

Corrected.

- page 13 "We have downloaded the structure files from ..." on what date?

Added.

- page 14 "Missing information is thus treated as information" suggest rephrasing this.

Missing information is thus treated as information, **in the sense that rather than attributing an imputed value a priori, the model's algorithm considers the missing value as another value of the specific predictor and as such is included as another node.**

- page 15. "pLDDT" might need to expand what that means.

Added.

- page 17. Figure 3. I don't find these sub-figures easy to interpret. Is there a better, simpler way to show the important information being conveyed here? The colours on the 3D protein models are particularly hard to see. I also find the Circos plots hard to understand, especially for MSH6.

We appreciate the concern of the Reviewer. Indeed, Fig. 3 underwent many versions during the manuscript's preparation. We have tried to use different visual representations, e.g. arc diagrams, but we agreed that the chord diagram was the best representation. As the other reviewers did not raise any concerns with the figure, we have decided not to change it.

Reviewer #2 (Remarks to the Author):

I think the reviewers have done a good job of and put a lot of effort into addressing my specific comments. The method looks interesting, the results seem promising, so I think it is worth publishing. I am somewhat skeptical that DeMAG really is superior to existing predictors, as it is always difficult to fully trust authors' evaluations of their own method. However, the best thing to do is publish the method and see how it performs in independent benchmarking.

We thank the Reviewer for his/her acknowledgement of the manuscript, which did improve with the Reviewer's comments. We share the concern regarding the difficulty of a fair benchmarking to unequivocally show a VEP's superior performance. We are curious to further validate DeMAG's performance. In addition to the webserver, we will offer DeMAG's prediction as a plugin for ENSEMBL VEP service so that hopefully more researchers will use the tool and decide on its usefulness.

Reviewer #3 (Remarks to the Author):

Authors have addressed all the comments raised in the initial review.

We thank the Reviewer for his/her previous comments that helped improving our manuscript.